# TGF-β Signaling

**DOI:** 10.3390/biom10030487

**Published:** 2020-03-23

**Authors:** Kalliopi Tzavlaki, Aristidis Moustakas

**Affiliations:** Department of Medical Biochemistry and Microbiology, Science for Life Laboratory, Uppsala University, Box 582, SE-751 23 Uppsala, Sweden; kalliopi.tzavlaki@imbim.uu.se

**Keywords:** extracellular matrix, phosphorylation, receptor serine/threonine kinase, signal transduction, SMAD, transcription, transforming growth factor-β, ubiquitylation

## Abstract

Transforming growth factor-β (TGF-β) represents an evolutionarily conserved family of secreted polypeptide factors that regulate many aspects of physiological embryogenesis and adult tissue homeostasis. The TGF-β family members are also involved in pathophysiological mechanisms that underlie many diseases. Although the family comprises many factors, which exhibit cell type-specific and developmental stage-dependent biological actions, they all signal via conserved signaling pathways. The signaling mechanisms of the TGF-β family are controlled at the extracellular level, where ligand secretion, deposition to the extracellular matrix and activation prior to signaling play important roles. At the plasma membrane level, TGF-βs associate with receptor kinases that mediate phosphorylation-dependent signaling to downstream mediators, mainly the SMAD proteins, and mediate oligomerization-dependent signaling to ubiquitin ligases and intracellular protein kinases. The interplay between SMADs and other signaling proteins mediate regulatory signals that control expression of target genes, RNA processing at multiple levels, mRNA translation and nuclear or cytoplasmic protein regulation. This article emphasizes signaling mechanisms and the importance of biochemical control in executing biological functions by the prototype member of the family, TGF-β.

## 1. Introduction

Biological signals regulate every aspect of physiological development of multicellular organisms and are important for the communication and coordination of cellular, tissue, and organ functions throughout life [1]. Among the large number of signaling molecules, the transforming growth factor β (TGF-β) family is highly conserved in the animal kingdom, and is thought to have appeared from the early days of multicellular (metazoan) evolution [2,3,4,5,6]. Across many species, the TGF-βs mediate a diverse range of embryonic and adult signaling functions that provide tissue-specific control of differentiation, proliferation, and cell-specific or tissue-specific motility [7,8,9,10].

The human TGF-β family includes thirty-three genes that encode for homodimeric or heterodimeric secreted cytokines [10,11]. These proteins are synthesized in a precursor form that is cleaved during processing through the secretory pathway and generate mature dimeric ligands most often held together via a single disulfide bond [11,12]. The family members have received a variety of names based on the history of their molecular identification, and include the activins, the bone morphogenetic proteins (BMPs), the growth differentiation factors (GDFs), the müllerian inhibiting substance (MIS), the nodal and the TGF-βs. Due to space limitation, this article will focus on the three TGF-βs (TGF-β1, -β2, -β3), which are collectively referred as TGF-β, and with occasional references to other family members.

Like in every other signaling network, regulation at multiple levels is of paramount importance so that the pathways operate physiologically and perform their normal function [1]. Genetic mutations can occur in several of the central molecular mediators of TGF-β signaling [8,9]. In addition, and more frequently, dedicated regulators of TGF-β family pathways malfunction, either due to their misexpression or due to genetic mutation, leading to weak or more often enhanced signaling input by the TGF-β signaling engine [8,9,10]. Such perturbations associate with the onset or alternatively with late stages of different diseases that include fibrotic disorders, chronic inflammatory conditions, and cancer [8,9]. In this article, the discussion will not focus on disease and only occasionally, examples from the pathophysiology of various diseases may be used, to illustrate control of signal transduction.

## 2. TGF-β Synthesis, Extracellular Deposition, and Activation

The elucidation of crystal structures for the various TGF-β family members, coupled to detailed biochemical and biophysical studies have shed light to the importance of TGF-β ligand processing, extracellular deposition and the mechanisms of their activation for presentation to signaling receptors [13,14]. We will now review this well-researched topic of TGF-β biogenesis and activation.

Similar to all other secreted proteins, TGF-β is synthesized by ribosomes attached to the rough endoplasmic reticulum of most cells, where removal of the short N-terminal signal peptide allows protein folding, glycosylation, and processing in subsequent biosynthetic steps during transport from the endoplasmic reticulum to the Golgi apparatus (Figure 1) [12,14]. TGF-β protein folding is intimately connected to the formation of intermolecular disulfide bonds, two in the N-terminal region that will later become the long prodomain and one in the C-terminal region that will later be the short mature ligand, resulting in obligatory dimerization of the unprocessed ligand (Figure 1) [12]. Glycosylation of the N-terminal part of the polypeptide is known to confirm latency, i.e., inactivity of the newly synthesized TGF-β [15], leading to the concept that functional activation is required at a later stage. Dimerization via disulfide linkage is followed by proteolytic cleavage of the polypeptides by furin family proteases, resulting in the formation of an N-terminal long dimeric and disulfide-linked propeptide, also known as latency-associated peptide (LAP), and a C-terminal short dimeric disulfide-linked polypeptide, also known as mature TGF-β (Figure 1) [10,12]. The two parts of TGF-β, generated after proteolytic cleavage, the LAP and the mature TGF-β, remain associated with each other and form the latent form of the ligand, often referred as large latent complex (LLC, Figure 1), whereby latency means lack of direct biological activity in the absence of further processing. Structural analysis of the latent form of TGF-β has elucidated the detailed molecular mechanism by which LAP directly covers the critical amino acids of the C-terminal dimer that are later used for interaction with the signaling receptors, and thus confers inactivation of the C-terminal dimer, when assembled as a latent complex [13]. It is of interest to consider the fact that during animal evolution, this principle of “masking” the mature dimeric ligand by the N-terminal part of the pro-peptide (LAP), in other members of the TGF-β family, has diverged so that the inhibitory polypeptides are actually expressed by a completely distinct gene. A good example of this difference is the inhibitory protein noggin that binds and inactivates a group of the BMPs [13].

Concomitant to the processing of the TGF-β polypeptide, crosslinking of the N-terminal LAP to other secreted proteins takes place (Figure 1). Two major families of proteins directly crosslink to the latent form of TGF-β in a cell type-specific or biological context-dependent manner; these are the latent TGF-β binding proteins (LTBPs) and the leucine rich repeat containing (LRRC) 32/33 proteins [14,16,17]. The LTBPs (such as LTBP1 and LTBP3 which interact with all three LAP-TGF-β1, -2, -3; LTBP4, which interacts with LAP-TGF-β1; LTBP2 which interacts with pro-myostatin/pro-GDF-8) are extracellular proteins, and upon secretion, mediate deposition of latent TGF-βs to the extracellular matrix (ECM) [14]. Via their ability to crosslink with additional proteins of the ECM, e.g., fibrillins and fibronectins, LTBPs provide the scaffolding units that tether latent TGF-βs to the ECM (Figure 1) [14]. The second group of latent TGF-β crosslinkers, the LRRC32/33 proteins are transmembrane proteins. LRRC32, also known as glycoprotein A repetitions predominant (GARP), is primarily expressed in immune cells, such as regulatory T cells (T_reg_) and crosslinks to latent TGF-β [16]. Similarly, LRRC33, primarily expressed in the plasma membrane of immune modulators of the brain, the microglial cells, also tethers latent TGF-β [17]. Whether incorporated into the complex ECM environment or associated with the plasma membrane of cells that regulate immunity, the disulfide-linked latent TGF-βs depend on mechanisms of activation and release of their mature dimeric ligands, as discussed below.

The structural details revealed by crystallography or analysis using nuclear magnetic resonance of the mature TGF-β ligands and their latent forms [13], provide exquisite examples of molecular specificity and evolution of growth factor domains that elegantly build a stereo-architecture that shields all the functional domains of the ligands, aiming at controlling carefully their mode of action, which awaits spatio-temporally controlled activation (Figure 1). Before discussing the mode of activation of the latent TGF-βs it is also worth presenting two more important points: a) latent or mature ligands in the TGF-β family are primarily known to exist as homodimers [13]. Yet evidence originally provided by the activin/inhibin sub-family, introduced the fact that heteromeric ligands exist, each one of them exhibiting different biological function relative to their homodimeric counterparts [13]. More recently, several other ligands of the TGF-β family have been shown to exert specific biological functions during vertebrate development in the form of heterodimers, examples being the BMP-2/BMP-7 and Nodal/GDF-1 heterodimers [18,19]. Thus, the concept of heterodimeric TGF-β family ligands requires further investigation. b) The structural studies have also shed light on the evolution of the TGF-β family and provide structural reasons that explain why the TGF-β LAP generates functional latency, whereas the equivalent N-terminal prodomains of other family members, e.g., BMP-9 do not [13].

Two key mechanisms have been uncovered that mediate activation of latent TGF-β and controversy exists about the relative contribution or significance of each mechanism (Figure 2). Early evidence suggested that the LAP portion of latent TGF-β was cleaved by ECM proteases such as matrix metalloproteases (MMPs) in cooperation with the tolloid-like family of proteases, an example of which is the protease known as BMP-1 (Figure 2) [20]. In the ECM microenvironment of fibroblasts, BMP-1 cleaves the LTBP1, causing release of the latent complex, which then promotes cleavage of LAP by MMP2 and eventual release of the mature TGF-β1 polypeptide [20]. The BMP-1-based mechanism is not unique to latent TGF-β1 activation but is also used for the activation of latent BMPs (BMP-2, BMP-4, BMP-11) and GDF-8/myostatin [21]. A theoretical difficulty encompassed in this model is that BMP-1/MMP activities would degrade specifically the LAP component, while preserving intact, by furthermore unleashing the activity of, the mature ligand (Figure 2). A second mechanism of latent TGF-β activation is based on the presence of an Arg, Gly, Asp (RGD) tri-peptide motif located near one end of the LAP dimer, which mediates direct interaction with integrin receptors tethered to the plasma membrane of cells that respond to TGF-β (Figure 2) [13]. Specific integrin receptors, i.e., α_v_β_6_ (in epithelial cells and fibroblasts) and α_v_β_8_ (in T_reg_) exhibit the potential to recognize the TGF-β1 LAP, leading to activation of mature TGF-β1 in the context of pulmonary fibrosis or immune suppression [16,22]. Recent structural analysis of this mechanism has revealed an exquisite process whereby, the integrin receptor complex, via its intracellular association with the actin cytoskeleton can exert force that distorts the folded structure of LAP, enforcing a mechanical release of the mature ligand from the well-designed cage generated by the TGF-β prodomain (Figure 2) [23]. This mechanism has also revealed that the integrin-mediated distortion of the TGF-β LAP that mediates mature ligand release generates an intermediate complex between LAP and mature TGF-β, that resembles the complex of pro-BMPs with mature BMP, which, as explained above does not present latency and is rather readily bioactive [23]. Whether, the integrin-based mechanism of latent TGF-β activation and the protease-dependent mechanism are coupled together (Figure 2), requires further investigation. This is plausible, as protease activity may primarily work on the ECM components, e.g., LTBPs, thus leading to a first intermediate with a more “accessible” LAP-mature TGF-β complex, to which integrin can then complete the activation process (Figure 2). Another exciting possibility worth scrutinizing further and via structural analysis of the relevant components is the delivery of active mature TGF-β directly on the surface of its signaling receptors or possibly on the surface of a coreceptor that then mediates ligand presentation to the signaling receptors.

## 3. Receptors for TGF-β Family Members

Subsequent to activation and release of the mature TGF-β from its latent form, direct association with receptors on the plasma membrane initiates the cascade of signal transduction that elicits biological actions on responding cells (Figure 2). All cell types examined so far carry receptors for the TGF-β family ligands in developing embryos and in young or adult animals, and these receptors can either signal via intrinsic catalytic activity or can act as coreceptors that either facilitate or prohibit ligand presentation to the signaling receptors [11]. The receptors that exhibit intrinsic catalytic activity are known to act as ATP-dependent protein kinases; they show specificity in phosphorylating primarily serine and threonine amino acids and with weaker efficiency they can phosphorylate tyrosines on substrate proteins, an experimentally tested fact that is compatible with the evolutionary placement of these receptor kinases in the dual specificity kinase branch of all human protein kinases [24]. TGF-βs themselves (TGF-β1, -β2, -β3) signal via a specific receptor complex made of two different proteins, the TGF-β type II receptor (TGFβRII) and the TGF-β type I receptor (TGFβRI, also known as activin receptor-like kinase 5, ALK-5), expressed in all cell types [11]. In endothelial cells, the TGF-β1/2/3 ligands can also engage with another type I receptor known as ActRL1/ALK-1 [25]. The two-receptor signaling system is obligatory as explained below, and is widespread in the TGF-β family. The variety of human cells can express five type II receptors and seven type I receptors, which, via oligomerization, generate signaling receptor complexes that serve all ligands in the extended TGF-β family. The structural features of these receptors are conserved through evolution and make the receptors members of the large family of type I transmembrane proteins. Their extracellular N-terminal part carries N-linked carbohydrate chains, specific subdomains that recognize the ligand, followed by the α-helical transmembrane domain. The C-terminal cytoplasmic domain is divided into a juxtamembrane domain that often plays regulatory roles, a protein kinase domain that accepts ATP and catalyzes substrate phosphorylation and a C-terminal tail that is mostly short, and in some receptors of the family can be extended much longer, providing additional regulatory functions [11].

The central mechanism of signal transduction by the TGF-β family receptors follows a well-characterized process of interactions and receptor-mediated phosphorylations (Figure 3). Accordingly, TGF-β associates first with a homodimeric TGFβRII, which acts as a high-affinity receptor, an interaction that causes a conformational adaptation between ligand and TGFβRII, in a manner that a new high-affinity binding site is formed for TGFβRI at the interface of ligand and TGFβRII [26]. Upon recruitment of two units of TGFβRI, the type II receptor kinase phosphorylates serine residues in the juxtamembrane subdomain of TGFβRI that is characterized by a short glycine- and serine-rich motif (GS), and thus, activates the type I receptor kinase [27]. Activation of TGFβRI depends on two interlinked events, first allosteric change in conformation of the receptor, which then leads to dissociation of the chaperone protein and negative regulator FKBP12 (FK506 binding protein of 12 kilodalton) from the type I receptor in a second step [28,29]. Thus, the biologically active receptor complex includes a dimeric ligand and a heterotetrameric receptor complex (Figure 2, Figure 3 and Figure 4). Upon activation, the TGFβRI phosphorylates its substrates; up until today, only a small family of proteins has been identified as type I receptor substrates, the SMAD family. Specifically, the type I receptor phosphorylates two different SMAD proteins in the case of TGF-β (and other family members such as activins and nodal), SMAD2 and SMAD3, or three different SMAD proteins in the case of BMPs ( also some GDFs and other ligand members), SMAD1, SMAD5, and SMAD8 [10]. This group of SMAD proteins that are substrates to the protein kinase of the type I receptors in the family, are collectively known as receptor-activated (R-) SMADs, as explained further later. The steps of type I receptor phosphorylation by the type II receptor and subsequent R-SMAD phosphorylation is the most central feature of the signaling mechanism by TGF-β family ligands (Figure 3). The phosphorylated R-SMADs will then transmit signals further downstream from the receptor complex (Figure 3), as described later in this article.

Alternative signaling pathways, involving the Ras and Rho GTPases and the mitogen activated protein (MAP) kinases were found to be activated by TGF-β since the early days of TGF-β signaling research [30,31]. Subsequent studies elaborated on the importance of such alternative signaling pathways and led to the concept of so-called non-SMAD signaling, which is explained in detail later in this article (see Section 7) [32].

The importance of TGF-β receptor oligomerization has been studied extensively. Receptor homodimerization seems to occur even in the absence of ligand and originates intracellularly in the endoplasmic reticulum as the receptors are post-translationally modified and prepared for deposition to the plasma membrane [33]. Such self-oligomerization applies to both type II and type I receptors of TGF-β, and it has also been extended to many other receptors in the family, including the BMP receptors [34]. As the TGF-β receptors do not form disulfide links to generate homooligomers, the process of dimerization is considered to depend on specific interactions based on motifs scattered along the cytoplasmic domain of these receptors [35,36]. Thus, ligand association with the receptors is thought to stabilize the homooligomers [34], whereas, ligand bound to the heterooligomer mediates the dissociation of the negative regulator and chaperone FKBP12, and can also induce a rotation of the receptors around their transmembrane domains, thus placing the intracellular domains in proper stereotactic configuration that facilitates the trans-phosphorylation reaction catalyzed by the TGFβRII on the GS domain serines of TGFβRI [37]. The details of the TGF-β ligand complex with the hetero-tetrameric receptor have been analyzed in depth by crystallographic studies [26,36,37]. An elegant genetic experiment utilized a point mutant TGF-β3 ligand in which one TGF-β3 subunit was normal and the other carried a point mutation resulting in loss of binding to the receptor complex [38]. This ligand exhibited reduced (25% to 50%) but not null signaling activity and structural analysis revealed that the wild type subunit formed proper complex with a hetero-dimeric receptor complex, confirming that the functional TGF-β receptor generates two interlinked but yet autonomously signaling receptor heterodimers [38]. Thus, receptor oligomerization is of great importance for transmission of signals by TGF-β, a process that is further regulated by the association of the signaling receptor with a variety of coreceptor proteins.

## 4. Coreceptors Facilitate or Inhibit Signaling via the TGF-β Receptors

Since the molecular cloning of the first TGF-β receptors, the concept of the coreceptor function, a receptor that bound ligand with high affinity but failed to signal based on its intrinsic catalytic activity, became evident [39,40]. The above refers to the type III receptor, also known as betaglycan due to its proteoglycan nature. Today, we appreciate a large variety of proteins that can act as coreceptors to the signaling TGF-β receptors; several of these proteins form complexes with one or both of the TGF-β receptor kinases and regulate signal transduction, yet they may or may not bind directly to TGF-β family ligands [41]. We will discuss here more extensively coreceptors that bind to TGF-β family ligands and will briefly refer to some examples of coreceptors that modulate signaling and provide molecular cross-talk with other signaling pathways to which the TGF-β coreceptor may play a more established function.

The transmembrane betaglycan or type III receptor carries chondroitin sulfate and heparan sulfate glycosaminoglycan (GAG) chains attached on two neighboring serine residues in a distinct subdomain of the extracellular domain of the protein; a large part of this ectodomain can be proteolytically cleaved and exists as an extracellular, soluble form of betaglycan [42]. Betaglycan binds with high affinity (low nM range) TGF-β1, -β2 and -β3, and promotes signaling by all these ligands as it can form complexes with the TGF-β receptor kinases [39,40,43]. The function of betaglycan as a positive coreceptor for TGF-βs has been highlighted in the case of the TGF-β2 isoform; TGF-β2 exhibits low affinity for the extracellular domain of TGFβRII [44], and thus, betaglycan mediates and promotes presentation of this ligand to the signaling receptor kinases, in an indispensable manner, as demonstrated in studies of mouse fibroblasts where the *TGFBR3* gene was knocked out [45]. Both SMAD and MAP kinase signaling downstream of the TGF-β receptor kinases is enhanced in cells that express betaglycan, as expected [46]. The cleaved, soluble betaglycan has the same affinity for TGF-β ligands as the membrane-bound coreceptor, and has been shown to act as a sink that secludes ligand from TGFβRII and thus negatively regulates signaling in normal cells [42] or in the context of metastatic breast cancer [47]. The long GAG chains are dispensable for TGF-β binding to the coreceptor [42,48], whereas under certain circumstances, the GAG chains may also play regulatory role and prohibit assembly of signaling TGF-β receptor kinases [49]. The embryonic lethality of mice with loss-of-function mutation of the *betaglycan* gene suggests a broad set of functions for this coreceptor that may exceed signaling by TGF-β [45]. In line with this hypothesis, betaglycan can bind other members of the TGF-β family, including the inhibin and certain BMPs, via distinct subdomains of its long extracellular part, thus playing coreceptor roles and promoting biological signaling by these growth factors [50,51]. Betaglycan also cooperates with the fibroblast growth factor (FGF) receptor 1 by mediating neuronal differentiation in brain tumor cells; this function is probably mediated via the heparan sulfate GAG chains of betaglycan that are known to bind to FGF family ligands [52].

A second TGF-β coreceptor with more tissue-restricted distribution is the disulfide-linked dimeric glycoprotein endoglin, whose name indicates predominant expression in endothelial cells [53]. Endoglin mediates its function towards signaling TGF-β family receptor kinases via both extracellular and via its short intracellular domain [54], and its role on TGF-β signaling is complex. In endothelial cells, TGF-β engages both its traditional TGFβRI and the ALK1 receptor as presented earlier [25]. The presence of endoglin seems to antagonize TGF-β signaling towards the TGFβRI and to rather promote signaling via the ALK1 receptor of the BMP branch [55]. The observation that TGFβRI can phosphorylate the short cytoplasmic tail of endoglin on two neighboring serine residues has been proposed as a mechanism that switches TGF-β signaling from TGFβRI to ALK1 in endothelial cells [56]. This mechanism is also compatible with studies on the two spliced forms of endoglin, the L (long form) that has the full-length cytoplasmic tail of 47 amino acids and the S (short form) that has only 14 amino acids in its C-terminal tail [57]. L-endoglin promotes the signaling switch from TGFβRI to ALK1/BMP SMAD, whereas the short S-endoglin, when expressed at sufficient levels, promotes strictly canonical TGF-β/TGFβRI signaling [57]. The complex roles of TGF-β and endoglin remain an active topic of research [41], which is stimulated by the need to explain the pathogenesis of hereditary hemorrhagic telangiectasia type I, in which endoglin is mutated and loses its function [58]. Other health disorders in which endoglin has been shown to play important roles are preeclampsia (PE) and pulmonary arterial hypertension (PAH) [59,60]. Similar to betaglycan, endoglin can be cleaved and release its extracellular domain to the extracellular microenvironment and to the circulation [59,60]. Studies focusing on the pathogenesis of PE and PAH, described that soluble endoglin binds TGF-β, even in the circulation, and thus inhibits its signaling activity, thus, at least in part, explaining the role of soluble endoglin in PE and PAH [59,60]. In addition to a TGF-β-based mode of action, soluble endoglin can associate with BMP-9; interestingly, careful stoichiometric analysis has revealed that soluble endoglin circulates in monomeric form in contrast to the disulfide-linked dimeric coreceptor on the plasma membrane [61]. The monomeric soluble endoglin in complex with BMP-9 was then shown to associate with membrane-bound dimeric endoglin and signal via receptor ALK1 in endothelial cells [61]. According to this model, PE can be explained by enhanced BMP-9 signaling [61], which may be compatible with the older model whereby soluble endoglin could bind and antagonize TGF-β in the vasculature of women developing PE or on the endothelial cells of the pulmonary artery in patients with PAH [59,60].

Membrane proteins anchored to the lipid bilayer via glycosylphosphatidylinositol (GPI) anchors regulate TGF-β signaling. Two such examples will be presented here. The protein Cripto (carries the more formal name epidermal growth factor-Cripto-1/FRL-1/Cryptic, EGF-CFC, based on its two major domains), was originally identified as a nodal coreceptor that enhances signaling via activin/nodal type I receptors (ALK-4, ALK-7) in early embryos [62], but was later shown to be expressed by cancer cells and bind TGF-β1, acting as a sink that limits availability of the ligand towards its signaling receptors [63]. Thus, Cripto, a coreceptor that is critical for normal embryogenesis, can also act oncogenically by suppressing physiological TGF-β signaling at the early onset of tumorigenesis when TGF-β suppresses cell proliferation and early hyperplastic growth [63]. A second GPI-linked coreceptor is the CD109 protein, which also binds directly TGF-β1, forms complexes with the signaling receptor kinases and acts as a negative factor in TGF-β signaling [64]. This negative function of CD109 relates to the process of receptor internalization and degradation discussed in a later section.

The BMP and activin membrane-bound inhibitor (BAMBI) directly associates with TGF-β and other ligands of the family, structurally resembles signaling TGF-β receptors, yet, due to the lack of a protein kinase domain intracellularly, it fails to signal but rather acts by interfering with signaling by the physiological receptor complexes [65]. TGF-β signaling transcriptionally induces the *BAMBI* gene [66], and BAMBI forms complexes with TGFβRI and SMAD7 (see below), thus blocking positive TGF-β receptor signaling to the SMAD pathway [67]. These mechanistic actions of BAMBI explain a large set of observations where misregulation of BAMBI expression associates with a variety of human diseases, including cancer, chronic inflammation, tissue fibrosis and cardiovascular disease [8,9].

A number of additional proteins that reside on the cell surface have been reported to form complexes with the signaling receptors for TGF-β, and thus contribute to modulation of signaling. Even nuclear proteins such as Ski and the mediator subunit MED12 have been reported to associate with and regulate TGF-β receptor activity. Authoritative accounts for many such proteins can be found in relatively recent articles, and we direct the reader to these articles for deeper or more comprehensive reading on this important topic [10,11]. Here we will only summarize general processes that govern TGF-β receptor activity.

Trans-phosphorylation of TGFβRI by TGFβRII is a central signaling event for the full activation of the receptor signaling capacity as described above [27]. Both TGF-β receptors carry additional phosphorylated residues, whereas the protein kinases mediating such events remain poorly characterized [10,11]. Counteracting phosphorylation, multiple protein phosphatases either completely remove the phosphates from target residues on the receptors, or provide quantitative reduction of the number of phosphorylated residues per receptor complex, an important concept that requires further study [10,11]. In addition, the signals and the specific phosphatases that get activated to dephosphorylate the TGF-β receptors remain open to future investigation. Unbiased phosphoproteomic analysis of the TGF-β receptors may shed light to this important open aspect of TGF-β signaling. Also, the generation of phospho-specific antibodies that can monitor the dynamics of phosphorylation and dephosphorylation and can probe the presence of active or inactive receptors in vivo in tissues is a major technical aspect missing from the analysis of this signaling pathway.

Examples of TGF-β receptor ubiquitylation and neddylation are described in this article. Sumoylation of the TGF-β receptor, the modification mediated by another member of the broad ubiquitin family, has also been described [10]. Equally important to the ubiquitin ligases are the actions of de-ubiquitylases (DUBs) that remove the ubiquitin modification from the receptors. In fact a growing list of DUBs and their mechanism of regulation during TGF-β signaling gradually becomes uncovered, a topic that has recently been reviewed [68].

Finally, the mechanisms of translational regulation, folding and glycosylation, including association with chaperone proteins like FKBP12, are poorly understood and deserve deeper analysis. An area of rapid growth is the post-transcriptional regulation of TGF-β receptor expression by non-coding RNAs. Most of these new examples focus on micro-RNAs that control mRNA stability and ribosomal translation of the receptor mRNAs, and impact on overall TGF-β signaling including SMAD pathway output [11].

## 5. TGF-β Receptor Internalization, Degradation, and Recycling

Upon ligand binding, it would be natural to assume that the heterotetrameric receptor complex is internalized via a classical receptor-mediated endocytic process. All evidence up to date would agree with this simple statement, yet the TGF-β receptor internalization studies have provided some interesting and complicated scenarios worth presenting here (Figure 4). In addition, the reader should consider seriously that in all experimental studies, the internalization of type II or type I receptors has been studied, yet it is never clear whether the internalized receptors initiated from the ligand-bound heterotetrameric receptor complex. Early studies on this topic using chimeric receptors where the extracellular domain of a receptor tyrosine kinase was fused to the transmembrane and cytoplasmic parts of each of the TGFβRII and TGFβRI, identified a steady flow of receptor internalization and recycling to the plasma membrane in the absence of ligand, whereas ligand-dependent internalization was directly linked to signaling activity, events that also presented distinct patterns in different cell types such as fibroblasts and epithelial cells [69]. Such differential trafficking of the TGF-β receptors can be in part explained by their presence in distinct plasma membrane domains, as exemplified by studies in polarized epithelial cells, where, the signaling receptor complexes have been mapped at lateral sites of the epithelial cells, where cell–cell contacts are made [70]. Independent evidence identified TGFβRI in association with the tight junction protein occludin in polarized epithelial cells [71], corroborating the evidence that the receptor complex may reside in distinct plasma membrane domains in different cell types, a fact that may also specify the pathway of receptor internalization.

The model of TGF-β signaling being associated with receptor endocytosis was also confirmed for native receptors [72]; receptor internalization after ligand stimulation was mapped to coincide with early endosomes enriched for the protein EEA1 (early endosomal antigen 1), and the adaptor protein SARA (SMAD anchor for receptor activation protein) (Figure 4). Furthermore, TGFβRI to R-SMAD signaling was mapped to early endocytic compartments, as discussed further later, whereas early complexes between TGF-β receptors and SMADs could form even on the cell surface prior to the formation of clathrin-coated pits and endocytosis to early endosome [73]. Deeper analysis into the internalization mechanism revealed two separate TGF-β receptor complexes on the plasma membrane (Figure 4): one localizing to clathrin-enriched coated pits and another localizing to cholesterol-enriched caveolae [74]. Interestingly, the receptor pool internalizing via clathrin-coated pits towards early endosomes marked by EEA1 and SARA protein localization was linked to SMAD signaling (described in Section 6), whereas, the caveolar receptor pool associated with the SMAD ubiquitin regulatory factor 2 (SMURF2), which mediated receptor polyubiquitylation and lysosomal degradation [74]. Partitioning of TGF-β receptors to caveolae is further supported by a specific physical interaction of TGFβRI with caveolin-1, a major protein constituent of the cholesterol-enriched caveolar membrane microdomains [75]. The GPI-linked TGF-β coreceptor CD109 discussed above, also regulates internalization, as this protein partitions in cholesterol-rich caveolar domains based on the interaction between CD109 and caveolin-1, and further promotes TGF-β receptor internalization via caveolae and final lysosomal degradation, thus mediating its negative action on TGF-β signaling [76]. In agreement with the caveolar pathway of TGF-β receptor downregulation, CD109 also promotes association of TGFβRI with SMAD7 and SMURF2 [77]. Whether caveolin-1 tethers CD109 and TGFβRI into a single complex or alternatively, these proteins interact with each other in a sequential manner, remains currently unknown. Whereas caveolin-1 and CD109 recruit TGF-β receptors to the caveolar compartment, the ubiquitin ligase c-Cbl (Casitas B-lineage lymphoma) acts on TGFβRII and neddylates (modifies by addition of NEDD-8 moieties) on two lysine residues of the cytoplasmic domain of the receptor, which enhances partitioning to clathrin-coated membrane domains and early endosomal internalization that is important for proper SMAD signaling [78]. Another regulatory mechanism that promotes endocytosis to the early endosome and SMAD signaling, while antagonizing SMAD7-mediated termination of signaling, involves the metalloprotease ADAM12 (a disintegrin and metalloproteinase 12), which associates with TGFβRII and in a protease-independent manner stabilizes the receptor and indirectly enriches endosomes with receptors that can signal downstream to R-SMAD phosphorylation [79].

Although most studies of TGF-β receptor internalization have focused on the signaling TGFβRII and TGFβRI (Figure 4), evidence focusing on the internalization of betaglycan confirms the above models for endocytosis. Specifically, betaglycan and TGFβRII (probably together with TGFβRI) reach the early endosome, assisted by the function of β-arrestin 2, a signaling regulator of G-coupled protein receptors [80]. TGFβRII was shown to phosphorylate the short cytoplasmic tail of betaglycan, leading to recruitment of β-arrestin 2 and internalization of the receptors with subsequent lysosomal degradation [80]. The studies focusing on betaglycan internalization have emphasized that TGF-β receptors can be degraded in lysosomes after internalization either via clathrin-coated pits or via caveolae [81].

TGF-β receptor recycling has been observed for both ligand-bound and free type II and type I receptors, which were shown to follow the well-established recycling pathway that generates recycling vesicles from the early endosomes to the plasma membrane, a pathway dependent on the catalytic activity of the Rab11 small GTPase (Figure 4) [82]. The endocytic protein Dab-2 (Disabled-2) that mediates sorting of clathrin-positive vesicles, participates in the recycling of TGFβRII, and its downregulation perturbs normal endosome formation and downstream SMAD signaling [83]. Furthermore, recycling of TGFβRII to the basolateral membrane of polarized epithelial cells depends on the retromer complex that ensures delivery to the correct membrane compartment from the endosome [84]. On the other hand, the ubiquitylation-dependent mechanism mediated by TRAF6 that activates MAP kinase signaling (see Section 7.2), is also responsible for the interaction of TGFβRI with the endocytic adaptor protein CIN85, causing receptor recycling to the plasma membrane via the Rab11-dependent trafficking pathway (Figure 4), and prolonged TGF-β signaling [85].

## 6. SMAD Signaling

As earlier mentioned in this article, SMAD proteins are the major effector molecules in the TGF-β signaling pathway (Figure 3). Upon ligand binding, and trans-phosphorylation by TGFβRII, TGFβRI activates SMAD2 and SMAD3 through phosphorylation at specific Ser residues in their C-terminal regions. These R-SMADs associate with the common mediator SMAD4 protein and form trimeric complexes, which are then shuttled to the nucleus (Figure 3). Nuclear SMAD complexes cooperate with DNA binding transcription factors but also with chromatin modifiers and can positively or negatively regulate the expression of TGF-β-responsive genes (Figure 3).

R-SMADs, as downstream effectors of TGF-β signaling were first identified in *Drosophila melanogaster* where they are known as Mad proteins (mothers against decapentaplegic, where decapentaplegic is the homolog of BMP-2/4 in *Drosophila*) [86], and in *Caenorhabditis elegans* where they were named as Sma proteins (based on the small body size of worms carrying mutations in the respective genes) [87]. It was later described that in humans, SMAD proteins were the mediators of TGF-β signaling [88], while also in the mouse mammary cell line NMuMG, it was demonstrated that TGF-β via the activation of SMAD proteins can promote epithelial to mesenchymal transition (EMT) [89]. Overall, SMAD signaling has been confirmed to mediate the biological effects of TGF-β based on in vivo studies of embryogenesis, adult tissue homeostasis and disease pathogenesis in a plethora of species, including *D. melanogaster*, *C. elegans*, *Xenopus laevis*, mouse and rat models. In humans, most of the studies have employed established cell lines, many from a variety of human tumors, and primary cells types of diverse, if not all, tissues.

### 6.1. Structure of SMADs

Both R-SMADs and co-SMAD proteins consist of two highly conserved domains, the Mad homology 1 (MH1) and the Mad homology 2 (MH2) domains (Figure 5). The MH1 domain, which is located at the N-terminus of the protein, contains nuclear localization signals (NLS) and a β-hairpin structure that mediates the binding of SMAD proteins on DNA [90]. The C-terminal MH2 domain of SMAD2 and SMAD3 contains an L3 loop, a motif that is critical for the SMAD–TGF-β receptor interaction [91,92], and the subsequent TGF-β-receptor induced phosphorylation of R-SMADs (Figure 5). This L3 loop is present also in the SMAD4 structure where it is essential for its interaction with R-SMADs during the formation of trimeric complexes [93]. A C-terminal Ser-X-Ser (SXS) motif on the MH2 domain of SMAD2 and SMAD3, serves as a phosphorylation site for TGF-β type I receptor, leading to their activation (Figure 5). This structural organization also extends to the BMP R-SMADs.

MH1 and MH2 domains are separated by the linker region, which is diverse among the different SMAD proteins and has loose secondary structure (Figure 5). The linker region contains multiple phosphorylation sites that are important for the regulation of the stability, subcellular localization of and activity of SMADs [94].

Alternatively spliced transcripts, encode a number of different isoforms of SMADs, which are known to have tissue-specific expression and to play significant and distinct roles during development [95].

### 6.2. Activation of SMADs

Upon ligand binding, the type II receptor phosphorylates the type I receptor, the latter dissociating from FKBP12 and being released from its inactive conformation, as explained in Section 3 [29]. The phosphorylation of the GS region on TGFβRI by TGFβRII, enhances the affinity of TGFβRI for the C-terminal SXS motif of R-SMADs [37]. The physical interaction between the R-SMADs and the type I receptor induces the phosphorylation of the two C-terminal serines on R-SMADs by the type I receptor (Ser^465^ and Ser^467^ in SMAD2 for example), an event that leads to conformational changes of R-SMADs and their subsequent dissociation from the receptor complex [96]. The specificity of the interaction between R-SMADs and type I receptor is determined by the L45 loop structural motif, a region in the type I receptor kinase domain, and the L3 loop on the R-SMAD MH2 domain [91,97]. The phosphorylation of both C-terminal serines is indispensable for the activation of R-SMADs, their dissociation from type I receptor, the formation of trimeric complexes with SMAD4 and the downstream propagation of signaling [98].

The endocytic protein SARA promotes the activation of R-SMADs as it is responsible for the recruitment of SMAD2 and SMAD3 to the TGF-β receptor (Figure 4). The cytoplasmic form of the promyelocytic leukemia (cPML) protein, which is a well-studied tumor suppressor protein in hematopoietic malignancies, stabilizes the complex between SARA and SMADs (Figure 4) [99]. Furthermore, recruitment and activation of SMADs depends also on Dab-2, an adaptor molecule that associates with SMADs and both type I and type II receptors thus promoting clathrin-mediated endocytosis of receptor complexes and transmission of TGF-β signaling (see Section 5, Figure 4) [100]. Upon their phosphorylation, the R-SMADs dissociate from the TGF-β receptor complex and interact with SMAD4 forming trimeric complexes that then translocate to the nucleus [101,102]. Hepatocyte growth factor-regulated tyrosine kinase substrate (Hgs) is a protein structurally similar to SARA that has also been described to interact with SMAD2 and promote its activation in a similar manner [103].

The conformational changes triggered by the phosphorylation of the Ser-X-Ser motif on R-SMAD monomers, not only drive the dissociation from the receptor complex but they also mediate the interaction between the phosphorylated C-terminus of an activated R-SMAD and the L3 loop on the MH2 domain of SMAD4 (or another R-SMAD), thus leading to the formation of SMAD oligomers [96,104,105]. SMAD trimeric complexes can be found in different compositions: as either oligomers consisting of three activated R-SMAD monomers (that can be either homo- or heterotrimers) or as oligomers composed of either two and one, or one and two phosphorylated R-SMADs, and SMAD4 respectively [104,106,107].

The fact that TGF-β can lead to phosphorylation of SMAD1 and SMAD5, the signal transducers of the closely related BMP pathway provides additional diversity and is essential for the complete TGF-β-induced transcriptional program to be elicited in certain cell types [108,109]. The coordinated activation of SMAD2/3 and SMAD1/5 may result in the formation of mixed R-SMAD complexes that target a unique subset of genes different than the genes regulated by non-mixed R-SMAD complexes [110].

### 6.3. Nucleocytoplasmic Shuttling of SMADs

Nuclear translocation of SMAD complexes is necessary in order to modulate gene transcription and the regulation of their subcellular distribution, is important for SMAD-mediated gene transcription (Figure 3). As aforementioned, all SMAD proteins contain NLS-like motifs in their MH1 domains (Figure 5), but the nucleocytoplasmic shuttling of each SMAD is regulated by distinct mechanisms. Activated SMAD3 interacts with importin-β1 via its MH1 domain and is subsequently imported into the nucleus in a Ran-dependent manner [111]. The long non-coding RNA NORAD promotes the nuclear translocation of SMAD complexes, as it is involved in the interaction between SMAD3 and importin-β1 [112]. On the other hand, the nuclear translocation of SMAD4 depends on its interaction with importin-α [113].

Nuclear export of SMADs is important for either the further propagation of signaling (via cycles of SMAD recycling) or for its termination. Exportin 4 recognizes a conserved sequence on the SMAD3 MH2 domain and carries the nuclear export of SMAD3 in a Ran GTPase-dependent manner [114]. RanBP3 directly interacts with dephosphorylated SMAD2 and SMAD3 and mediates their nuclear export, which results in the termination of signaling [115].

Moreover, nucleocytoplasmic shuttling of SMAD2, SMAD3, and SMAD4 is also regulated by their direct interaction with the nuclear pore proteins CAN/Nup214 and Nup153 [116,117]. Subcellular localization of R-SMADs is also modulated by the motor protein dynein light chain km23-1, which interacts with SMAD2 and is crucial for the nuclear translocation of activated SMAD2 and thus for SMAD2-mediated transcription [118]. Similarly, dynein light chain km23-2, associates with SMAD3 and promotes SMAD3-dependent transcription [119].

### 6.4. Posttranslational Modifications of SMADs

Posttranslational modifications of SMAD proteins control SMAD stability and function, providing an extra level of regulation of the physiological responses to TGF-β. Phosphorylation, dephosphorylation, ubiquitylation, sumoylation, and other modifications occur in response to a variety of growth factors and other signaling cascades. [94].

MH1 and MH2 domains, but also the less conserved linker regions of SMADs, contain sites of phosphorylation for various kinases such as MAP kinases and cyclin-dependent kinases (CDKs), that modulate SMAD activity, by for example generating a phosphorylated residue platform for recruitment of ubiquitin ligases [94]. SMAD2 can be phosphorylated by extracellular signal-regulated kinase 1 (ERK1) at Thr^8^ on the N-terminus (MH1 domain) and this modification leads to increased SMAD2 stability, which eventually results to enhanced SMAD-dependent transcription [120]. SMAD3 contains phosphorylation sites for protein kinase C (PKC) on the MH1 domain (Ser^47^ and Ser^70^). This phosphorylation abrogates the direct DNA binding of SMAD3, thus inhibiting SMAD3-dependent transcription [121]. The cGMP-dependent protein kinase 1 (PKG-1) targets SMAD2 and SMAD3 for phosphorylation in order to promote proteasomal degradation of activated R-SMADs [122], while casein kinase 1 gamma 2 (CKIγ2) was identified as another negative regulator of activated SMAD3 stability as it promotes its ubiquitylation and degradation through phosphorylation at Ser^418^ on the MH2 domain [123]. The p21-activated kinase 2 (PAK2)-dependent phosphorylation of SMAD2 at Ser^417^ close to the L3 loop, interferes with the TGFβRI-SMAD2 interaction and attenuates SMAD2 activation [124]. The common mediator SMAD4 is a target of ERK, which promotes nuclear accumulation and thus, enhanced transcriptional activity of SMAD4 by phosphorylating Thr^276^ on the SMAD4 linker region [125]. On the other hand, liver kinase B1 (LKB1) negatively regulates SMAD4-dependent transcription through phosphorylation at Thr^77^ within the β-hairpin of the SMAD4 MH1 domain, which prevents SMAD4 from binding to DNA [126]. The murine protein serine/threonine kinase 38 (MPK38) was also described to phosphorylate SMAD2, SMAD3 at their linker region (Ser^245^ and Ser^204^ respectively) and SMAD4 at Ser^343^ on the MH2 domain, enhancing TGF-β signaling [127].

The C-terminal activating phosphorylation and the inhibitory phosphorylation at the linker region of SMADs can be reversed by the action of different phosphatases. The dephosphorylation of C-terminal serines leads to the deactivation of R-SMADs and the attenuation of TGF-β signaling. Protein phosphatase Mg^2+^/Mn^2+^ dependent 1A (PPM1A/PP2CA) dephosphorylates the C-terminus of R-SMADs and promotes their nuclear export [128], while myotubularin related protein 4 (MTMR4), a dual-specificity protein phosphatase, dephosphorylates and deactivates R-SMADs in the early endosomes [129]. In contrast, small C-terminal domain phosphatases 1, 2, and 3 (SCP1, -2, and -3) remove the inhibitory phosphorylation from the linker region of R-SMADs and enhance TGF-β signaling [130,131].

SMAD steady state levels and activity are also regulated by ubiquitylation, whereby one or more ubiquitin molecules are covalently attached to proteins. Several E3 ubiquitin ligases target SMADs for proteasome-mediated degradation, among them SMURF1 and SMURF2, that are the most extensively studied HECT domain E3 ligases that ubiquitylate SMAD proteins. SMURF2 polyubiquitylates and targets for degradation SMAD2 [132], while it induces multiple mono-ubiquitylation of SMAD3, thus inhibiting the formation of SMAD3 complexes [133]. Neural precursor cell expressed, developmentally downregulated 4-2 (NEDD4-2/NEDD4L), another member of the HECT family of E3 ligases, recognizes SMAD2 and SMAD3 once they are phosphorylated at the linker region by nuclear CDK8/9 and ubiquitylates them, promoting their degradation [134,135]. SMAD3 levels are also regulated by the E3 ligase C terminus of HSC70-interacting protein (CHIP) [136], but also by the E3 ligase complex SCF (Skp1, Cullin1 and Fbw1a)/ROC that interacts with the activated SMAD3, promotes its nuclear export and induces its degradation [137]. Additionally, non-activated SMAD3 is recognized by Axin/glycogen synthase kinase 3 beta (GSK3β) complex, where Axin facilitates the phosphorylation of SMAD3 at Thr^66^ on the MH1 domain by GSK3β, which triggers SMAD3 ubiquitylation and degradation [138]. Ubiquitylation of SMAD2 by Itch E3 ligase facilitates the interaction between SMAD2 and the TGF-β receptor and positively regulates TGF-β-dependent transcription [139]. Interestingly, the ubiquitylation of activated R-SMADs by the nuclear RING-domain E3 ligase Arkadia, strongly enhances their transcriptional activity while at the same time promotes their degradation, providing an interesting mechanism that ensures efficient regulation of target genes followed by termination of the signaling at the end of the cascade [140]. The common mediator SMAD4 is a target of mono-ubiquitylation at its MH2 domain and this is linked to enhanced oligomer formation with R-SMADs and subsequent TGF-β-induced transcription [141], while SMAD4 ubiquitylation by the SCF^skp2^ E3 ligase complex, promotes its degradation [142]. The tripartite motif-containing 33 (TRIM33) E3 ligase (also known as transcriptional intermediary factor 1 gamma (TIF1γ) or Ectodermin), also triggers SMAD4 mono-ubiquitylation to promote disruption of SMAD complexes [143,144]. SMAD ubiquitylation can be reversed by the function of DUB enzymes. Ubiquitin carboxyl-terminal hydrolase 15 (USP15) has been identified as a DUB that opposes R-SMAD monoubiquitylation and allows the binding of SMADs on regulatory DNA sequences [145], while SMAD4 monoubiquitylation at Lys^519^ that inhibits SMAD4 association with R-SMADs is reversed by the function of ubiquitin-specific protease 9x (USP9X/FAM) [146].

During sumoylation, an enzymatic process similar to ubiquitylation, small ubiquitin-like modifier proteins (SUMOs) are covalently attached to protein substrates and modify their function, their subcellular localization or their interactions with other proteins. Protein inhibitor of activated STAT1 (PIAS1), a SUMO E3 ligase, has been described to promote SMAD4 sumoylation, leading to further enhanced TGF-β-induced transcription [147]. Conversely, another member of the PIAS protein family, the PIASy, inhibits TGF-β signaling by sumoylating SMAD3 and inhibiting SMAD3-mediated transcription [148].

TGF-β promotes the acetylation of SMAD2 and SMAD3 by the histone acetyltransferase p300, and this modification enhances the transactivation activity of R-SMADs [149].

Additionally, SMAD3 and SMAD4 are also substrates for poly(ADP-ribose) polymerase-1 (PARP-1), which poly-ADP-ribosylates them on the MH1 domain, promotes their dissociation from the DNA and thus, attenuates SMAD-dependent transcription [150]. These diverse mechanisms of posttranslational modification of SMADs can regulate TGF-β response of specific, but not all, groups of genes, the importance of which need to be further clarified.

### 6.5. SMADs in Transcription

Trimeric nuclear SMAD complexes regulate the expression of target genes (Figure 3). SMAD3 and SMAD4 can directly bind on specific DNA sequences (with low affinity though) that have been characterized as SMAD-binding elements (SBEs). In particular, SMAD3 and SMAD4 can recognize and bind to half of the palindromic octamer 5′-GTCTAGAC-3′ via a β-hairpin in their MH1 domain that embeds itself in the major groove of the DNA octamer [90,151,152]. Moreover, it was more recently described that this β-hairpin structure is able to also bind GC-rich regulatory elements, and more precisely the 5′-GGC(GC)|(CG)-3′ consensus sequence [153]. In contrast, it had been demonstrated that the most prevalent isoform of SMAD2 does not bind directly on to DNA because of a unique 30 amino acid sequence (E3 insert) in its MH1 domain, absent from the MH1 domain of SMAD3 and SMAD4 (Figure 5), which interferes with its DNA-binding activity [151]. Actually, the DNA-binding capacity of SMAD2 depends on the conformation adopted by the E3 insert (Figure 5). An open conformation allows SMAD2 to contact DNA while a closed one hinders the β-hairpin structure from interacting with the DNA [154]. Interestingly, it was recently described that this intrinsic structural difference between the two proteins, plays also a role in their subcellular distribution as SMAD3 appears to be mostly nuclear in the absence of ligand while SMAD2 is cytoplasmic and thus becomes more efficiently activated upon TGF-β stimulation [155].

However, even though SMAD complexes themselves have an intrinsic low DNA-binding affinity, they interact with a wide variety of DNA-binding transcription factors, chromatin modifiers and other coregulators in order to efficiently control the expression of target genes. Many of these SMAD-interacting nuclear partners are cell-type specific transcription factors that can direct SMAD complexes to specific promoters and thus determine the context-dependent effects of TGF-β signaling [156].

Histone acetyltransferases (HATs) and other chromatin modifiers are involved in the regulation of SMAD-dependent transcription either by modifying histones or/and by regulating SMAD activity. SMAD4 acts as transcriptional coactivator as it stabilizes the ligand-dependent interaction of SMAD3 with the histone acetyltransferases CREB-binding protein (CBP) and p300, which leads to enhanced SMAD3 transcriptional activity [157]. SMAD3-mediated transcription is also potentiated by the interaction of SMAD3 with the histone acetyltransferase p300/CBP-associated factor (P/CAF). P/CAF can enhance the transactivation of SMAD3 independently or in cooperation with p300 and CBP [158]. Similarly to P/CAF, general control of amino acid synthesis protein 5-like 2 (GCN5) is another HAT that acts as a coactivator for R-SMADs and further enhances TGF-β-induced transcription [159]. The Histone 3 (H3) acetylation by SMAD2-dependent recruitment of p300 and switch/sucrose non-fermentable (SWI/SNF) remodeling complex on SMAD2-dependent promoters, suggests that chromatin remodeling is essential for SMAD-mediated transcription [160]. The TGF-β-induced recruitment of SET domain bifurcated histone lysine methyltransferase 1 (SETDB1/ESET), which mediates specifically H3 Lys^9^ methylation, by SMAD3 to the regulatory sequences of the *SNAIL* gene, leads to attenuation of TGF-β-induced *SNAIL* expression [161]. This mechanism of fine-tuning TGF-β-dependent transcription provides a balance between histone modifications with opposing functions (e.g., the activating mark H3 Lys^9^ acetylation versus the repressive mark H3 Lys^9^ trimethylation). Whether such balanced or sequential modification of histones is a prerequisite for target gene regulation by TGF-β signaling, remains to be examined.

The zinc finger protein 451 (ZNF451) represses SMAD-mediated transcription by blocking the recruitment of p300 by SMADs, which leads to reduced H3 Lys^9^ acetylation of the promoters of target genes [162]. The SMAD-interacting protein TRIM33 is a histone-binding protein (a ”reader” of specific histone modifications) that also has E3 ubiquitin ligase activity [144]. The recruitment of TRIM33 to chromatin depends on SMAD4 and after binding of SMAD2/3-TRIM33 complexes on the regulatory sequences of target genes, further histone modifications take place that switch the state of chromatin from poised to active [163]. At the same time, upon histone binding, the E3 ligase activity of TRIM33 is induced, resulting in DNA-bound SMAD4 mono-ubiquitylation, subsequent disruption of SMAD complexes and release from SMAD-dependent promoters [143]. This probably acts as an intrinsic negative feedback mechanism that restricts the time SMAD complexes are bound on the promoters of target genes and positively regulate transcription.

### 6.6. SMADs in Posttranscriptional Regulation

SMADs can regulate gene expression also at the posttranscriptional level, affecting mRNA splicing, stability and translation via RNA-binding proteins (RBPs) and non-coding RNAs (ncRNAs).

Once phosphorylated at Thr^179^, SMAD3 interacts with the RNA-binding protein poly(rC)-binding protein 1 (PCBP1) and promotes alternative splicing of the cluster of differentiation 44 (CD44) pre-mRNA, in favor of the specific isoform CD44s, which is important for TGF-β to induce EMT [164]. SMADs can also control the maturation process of miRNAs. They bind on conserved sequences on the primary miRNA transcripts (pri-miRNAs) and facilitate recruitment of the DROSHA microprocessor complex (DROSHA/DGCR8/RNA helicase p68) in a ligand-dependent manner, thus promoting efficient cleavage of a subset of pri-miRNAs to pre-miRNAs (precursor miRNAs). SMADs bind to the double-stranded stem of folded pri-miRNAs, using the same recognition mechanism via their β-hairpin, which associates with the SBE motif but on pri-miRNA this time [165,166]. Additionally, TGF-β/SMAD signaling can regulate the expression of several long non-coding RNAs (lncRNAs) that function as mediators of TGF-β responses [167,168]. Finally, R-SMADs regulate signaling posttranscriptionally by participating in another negative feedback mechanism. They associate with the m^6^A methyltransferase complex METTL3-METTL4-WTAP, which is recruited onto nascent transcripts and mediates N^6^-adenosine methylation on the RNA. This m^6^A methylation destabilizes the transcripts and leads to their degradation [169].

### 6.7. Function and Regulation of Inhibitory SMADs

Initiation and propagation of TGF-β signaling is counteracted by the activity of SMAD6 and SMAD7, which are the inhibitory SMADs (I-SMADs) (Figure 3). In terms of structure, I-SMADs share homology with the R-SMADs and the co-SMAD at the MH2 domain, although they lack the SXS motif, whereas their amino-terminal regions (N-domains) are not only significantly different from those of other SMADs, but they are partially conserved in between SMAD6 and SMAD7, and confer rudimentary parts of the MH1 domain (Figure 5).

I-SMADs, via their MH2 domain, physically associate with the TGF-β type I receptor and antagonize TGF-β signaling by inhibiting the TGF-β-induced phosphorylation and activation of R-SMADs. [170,171,172]. The N-domain of SMAD7 enhances its inhibitory activity by facilitating the interaction of the SMAD7 MH2 domain with the TGF-β receptor, making SMAD7 a more potent inhibitor of TGF-β signaling compared to SMAD6, which preferentially inhibits BMP signaling [173,174]. Besides its inhibitory activity on the phosphorylation of R-SMADs, SMAD7 antagonizes the TGF-β pathway by recruiting the E3 ubiquitin ligases SMURF1 and SMURF2 to the type I receptor and promoting its ubiquitylation and subsequent degradation [170,173,175]. SMAD7 interacts with many members of the HECT-domain ubiquitin ligases; in addition to SMURF1 and SMURF2, the family members WWP1 (WW domain-containing protein 1) and NEDD4-2 act in a similar manner to the SMURFs, enhance cytoplasmic accumulation of SMAD7, its association with TGFβRI, polyubiquitylation and degradation of the receptor and even degradation of the R-SMADs as is the case for NEDD4-2 [176,177]. TGF-β promotes the cytoplasmic accumulation of the predominantly nuclear, in the absence of ligand, SMAD7, and also induces SMAD7 mRNA expression, thus creating a negative feedback loop that tightly controls signaling [172,178]. As a balance to the negative impact of I-SMADs on TGF-β signaling, TGF-β induces the expression of transforming growth factor-β-stimulated clone-22 (TSC-22), which competes with SMAD7 for interacting with the TGF-β type I receptor, protecting it from degradation [179].

The nuclear pool of I-SMADs provides another level of regulation for TGF-β signaling. Nuclear SMAD7 can function as an antagonist for TGF-β signaling as it can compete with functional SMAD complexes for binding on the regulatory sequences of TGF-β target genes [180]. Moreover, the physical interaction of SMAD6 with histone deacetylases (HDACs), is an indication of their potential role as transcriptional regulators for target genes [181].

Similarly to R-SMADs, inhibitory SMADs are regulated via posttranslational modifications. MPK38-mediated phosphorylation of SMAD7 at Thr^96^ promotes SMAD7 translocation to the cytoplasm but also interferes with the association of SMAD7 with TGFβRI and thus has a positive effect on TGF-β signaling [127]. SMAD7 can also be phosphorylated at Ser^249^ (the responsible kinase has not yet been identified) and this modification affects the transcriptional activity of SMAD7 independently of TGF-β signaling [182]. Nuclear SMAD7 can be acetylated by acetyltransferase p300 and deacetylated by HDAC1 and NAD-dependent protein deacetylase sirtuin-1 (SIRT1). The acetylated SMAD7 is more stable while its deacetylation promotes its ubiquitylation and degradation [183,184,185]. Non-acetylated SMAD7 is also a substrate for the SET9 methyltransferase. SMAD7 methylation promotes its association with the E3 ubiquitin ligase Arkadia, enhancing its ubiquitylation and subsequent degradation [186]. Thus, I-SMADs may have evolved in order to provide specific negative control on the R-SMAD/co-SMAD complex that transmits the positive signal by TGF-β.

## 7. Non-SMAD TGF-β Signaling Pathways

In addition to the canonical SMAD signaling, TGF-β can regulate downstream cellular responses also via other signal transducers (Figure 6) in a context-dependent manner.

### 7.1. ERK MAP Kinase Pathway

The initial observation that TGF-β rapidly activates Ras proteins in epithelial cells came before the characterization of the TGF-β/SMAD pathway [30,187], and later, TGF-β-dependent activation of ERK MAP kinase signaling was described [31]. One mechanism of activation of the Ras-MAP kinase pathway by TGF-β receptors depends on the weak, yet detectable catalytic activity of TGFβRI kinase towards tyrosines [188]. Thus, it was described in Mv1Lu mink lung epithelial cells and 3T3-Swiss mouse fibroblasts, that when TGFβRI phosphorylates the adaptor protein Shc on tyrosine, it initiates a docking site for downstream signaling mediators, Grb2, Sos, which then activate the Ras GTPase, leading to the sequential activation of c-Raf, MEK (MAP kinase/ERK kinase) and ERK1/2 kinases (Figure 6A) [188]. Interestingly, in human keratinocytes (HaCaT), it was demonstrated that ERK activation occurs when TGF-β receptor complexes are localized in cholesterol-rich lipid rafts [189], while SMAD activation is induced by the clathrin-dependent endocytosis of TGF-β receptor complexes (as described in more detail in Section 5) [74].

### 7.2. JNK and p38 MAP Kinase Pathways (via TAK1)

c-Jun N-terminal kinase (JNK) and p38 MAP kinase pathways (Figure 6B) are important for the regulation of different cellular processes such as inflammation, cell differentiation and apoptosis.

TGF-β can rapidly activate JNK and p38 in a SMAD-independent manner via activation of MAP kinase kinases (MKKs). MKK4 is upstream of JNK while MKK3 and MKK6 activate p38 [190,191,192]. The MKKs are themselves substrates for the MAP kinase kinase kinases and among them, the mitogen-activated protein kinase kinase kinase 7 (MAP3K7), also known as TGF-β-activated kinase 1 (TAK1), was found in epithelial cells to be activated in response to TGF-β via TGF-β activated kinase 1 binding protein 1 (TAB1) phosphorylation [193]. Upon ligand binding, TGF-β receptor complexes interact with the tumor necrosis factor receptor associated factor 6 (TRAF6) E3 ubiquitin ligase, inducing TRAF6 activation via autoubiquitylation and promoting association of TRAF6 with TAK1. Activated TRAF6 can then activate TAK1 via Lys^63^-linked polyubiquitylation, a crucial step for the subsequent activation of the p38 MAP kinase pathway [194,195,196]. The TGF-β receptor to TRAF6 mechanism has been proposed to be independent from the kinase activity of TGFβRI [194,197], which emphasizes the important function of ligand-mediated receptor oligomers (Figure 6B). TRAF4 is another E3 ligase described to interact with the TGF-β receptor complex in a ligand-dependent manner in order to trigger TAK1 Lys^63^-linked polyubiquitylation (Figure 6C). However, compared to TRAF6, TRAF4 can also enhance SMAD-signaling by stabilizing TGF-β receptor levels [198]. TRAF6-mediated polyubiquitylation of TAK1 is also required for efficient activation of nuclear factor kappa-light-chain-enhancer of activated B cells (NF-κB) signaling (Figure 6B), thus mediating a cross-talk between TGF-β and NF-κB pathways [199].

Besides its role as an activator of p38 signaling downstream of TGF-β, TAK1 promotes also R-SMAD phosphorylation at the linker region in craniofacial neural crest-derived mesenchymal cells, acting as a negative regulator of canonical TGF-β signaling as well [200].

Interestingly, inhibitory SMADs appear to have significant roles in the regulation of p38 MAP kinase signaling (Figure 6B). SMAD7 has been described to act as a scaffolding protein for p38 and its upstream kinases MKK3 and TAK1, thus facilitating the activation of the pathway in prostate cancer cells [201]. In contrast, it was demonstrated in AML-12 liver cells and primary hepatocytes, that SMAD6 can negatively regulate p38/JNK signaling by recruiting the deubiquitylase A20 in order to inhibit TGF-β-induced TRAF6 Lys^63^-linked polyuibiquitylation [202].

### 7.3. PI3K-AKT Pathway

Phosphatidylinositol 3-kinase/protein kinase B (PI3K/AKT) signaling is activated by TGF-β via the direct interaction of p85, the regulatory subunit of PI3K with the TGF-β receptor complex (Figure 6D). In epithelial cells, interaction of p85 with the type II receptor appears to be constitutive while the interaction with the type I receptor is induced by TGF-β [203]. For AKT activation to occur, its Lys^63^-linked ubiquitylation by TRAF6 is important as it results to the recruitment of AKT to the plasma membrane and to its activating phosphorylation [204]. Actually, TRAF6 also triggers Lys^63^-linked ubiquitylation of the p85 subunit of PI3K in a TGF-β-dependent manner, and this leads to the activation of PI3K in prostate cancer cells, the production of phosphatidylinositol-(3,4,5)-trisphosphate and the recruitment of AKT to the plasma membrane [197]. Additionally, it was described that via PI3K, TGF-β can induce mammalian target of rapamycin complex 2 (mTORC2), which can in turn also phosphorylate and activate AKT (Figure 6D), thus contributing to TGF-β-induced EMT and cell invasion in mouse mammary NMuMG cells [205]. Furthermore, AKT activation leads to increased stabilization of the EMT transcription factor SNAIL, thus promoting TNF-α-induced EMT [206]. Finally, it has been demonstrated in hepatocytes, that initiation of TGF-β-induced AKT signaling is dependent on dynamin-mediated endocytosis via the caveolar pathway [207], even though this is a well-described route leading to the degradation of the TGF-β receptors [74].

### 7.4. TGF-β Type I Receptor Intracellular Domain Signaling

TGF-β receptor internalization (Figure 4) has also been intimately linked to proteolytic processing of the receptors and generation of a more direct signaling pathway (Figure 6E). TNF-α converting enzyme (TACE), also known as ADAM17, cleaves the extracellular domain of TGFβRI, which is released extracellularly and results in downregulation of TGF-β receptor signaling as it was demonstrated in HaCaT keratinocytes and in breast cancer cells [208]. During TGF-β signaling, TACE/ADAM17 remains inhibited by association with the inhibitory protein TIMP3 (tissue inhibitor of metallopeptidase 3); activation of MAP kinase (ERK or p38) signaling removes TIMP3 from the dimeric TACE and promotes monomeric and bioactive TACE, which can then process TGFβRI and terminate the signal by the TGF-β receptor [209]. This mechanism, established in breast cancer cells, links MAP kinase signaling to the TGF-β receptor downregulation as a possible negative feedback mechanism (Figure 6E). This is consistent with the evidence that activation of the TRAF6-mediated polyubiquitylation of TGFβRI that is induced by TGF-β when the receptor oligomeric complex forms, is required for the cleavage of the receptor by TACE/ADAM17 [210]. A single lysine residue (Lys^178^) in the intracellular domain of TGFβRI becomes polyubiquitylated by Lys^63^-interlinked ubiquitin polymers, and promotes receptor cleavage [211], possibly via a conformational change on the receptor structure or via recruitment of intermediate ubiquitin-binding adaptor proteins. Intermediate to TACE/ADAM17 activation, in addition to MAP kinases, is also the activity of protein kinase C ζ (PKCζ) [210]. In parallel, in prostate cancer cells, TRAF6 can polyubiquitylate the membrane-embedded protease presenilin-1 (PS-1), which performs a second proteolytic cleavage on TGFβRI, causing the release of the complete intracellular domain (ICD) of the receptor, and stimulating the translocation of the TGFβRI ICD to the nucleus [212]. In the nucleus, the TGFβRI ICD has been shown to associate with chromatin and with transcriptional regulators thus generating a more direct signal (compared to SMAD or MAP kinase signaling) for gene expression regulation downstream of TGF-β (Figure 6E) [212]. This process clearly illustrates that TGF-β receptor downregulation and degradation is intimately linked to generation of positive signaling from the receptors to the gene (Figure 4 and Figure 6E).

### 7.5. JAK-STAT Pathway

Janus kinase (JAK)—signal transducers and activators of transcription (STAT) signaling can be activated by TGF-β in a context-dependent manner. In fibroblasts, the JAK-STAT pathway acts as mediator of the TGF-β profibrotic effect. In fact, TGF-β promotes the phosphorylation of JAK2, which leads to the phosphorylation and activation of STAT3 (Figure 6F) [213]. In hepatic cells, it has been described that JAK1 constitutively interacts with the TGF-β type I receptor and induces STAT3 phosphorylation upon TGF-β stimulation in a SMAD-independent manner [214]. Activation of the JAK1-STAT3 pathway appears to be important for the regulation of a subset of TGF-β target genes. Furthermore, it has been described that cooperation of STAT3 with SMAD3 is required for TGF-β fibrogenic responses in hepatic stellate cells [214].

### 7.6. Rho-(like) GTPase Pathway

Rho- and Rho-like GTPases are important regulators of cytoskeletal organization and cell motility and their TGF-β-induced activation contributes to TGF-β-induced EMT. TGF-β rapidly induces activation of RhoA and Cdc42 GTPases in epithelial cells in order to promote rapid actin reorganization and membrane ruffling, while for long-term responses (such as stress fiber formation), cooperation of Rho-GTPases with SMAD signaling is required (Figure 6G) [215,216]. Unlike the rapid activation of RhoA that is not dependent on activation of SMADs, its late activation by TGF-β is SMAD-dependent as SMAD signaling induces the expression of guanine exchange factor NET1 which in turn activates RhoA to promote stress fiber formation [217]. As a negative feedback mechanism, prolonged TGF-β stimulation leads to downregulation of NET1 via the induction of the miRNA *miR-24*, which targets the transcript of a specific NET1 isoform and blocks its translation [218].

Rho-GTPase signaling is negatively regulated by TGF-β via Par6, a regulator of epithelial cell polarity. Upon ligand binding and TGF-β receptor complex formation, Par6, which is found associated with the TGFβRI, gets phosphorylated by the TGFβRII (Figure 6G). This leads to the recruitment of SMURF1 ubiquitin ligase, which polyubiquitylates and marks for degradation the RhoA GTPase [219]. This results in the rapid loss of tight junctions in epithelial cells, an early event of the EMT process.

In fibroblasts, a different cell type, TGF-β can also activate PAK2 via Rac1 and Cdc42 Rho-like GTPases (Figure 6G), to promote fibroblast morphological transformation [220].

### 7.7. Other TGF-β Activated SMAD-Independent Pathways

It has been reported that TGF-β rapidly promotes activation of protein kinase A (PKA) in mesangial cells, and the activated PKA then contributes to TGF-β-induced cAMP response element-binding protein (CREB) phosphorylation and *fibronectin* (*FN1*) gene transcription [221]. However, it was later demonstrated that TGF-β activates PKA in a SMAD-dependent manner, as the SMAD3/SMAD4 complex directly interacts with the regulatory subunit of PKA, thus allowing the catalytic subunit of the kinase to exert its function [222].

c-Abl tyrosine kinase is another direct target of TGF-β signaling and TGF-β-induced activation of c-Abl is involved in TGF-β-mediated fibrosis [223].

Finally, time-dependent analysis of phosphoproteome and proteome changes that human keratinocytes undergo in response to TGF-β, provided evidence that the phosphorylation status of many proteins changes with rapid kinetics, suggesting that SMADs may not be involved [224].

Overall, the relative impact of SMAD-dependent (Figure 3) and non-SMAD signaling (Figure 6) for the specification of various cellular processes awaits further investigation.

## 8. Signaling Cross-Talk

TGF-β signaling outcomes depend also on cross-talk of the TGF-β pathway with other signaling pathways as the function of TGF-β receptors and SMADs is modulated by other signaling effectors. Additionally, among the SMAD-interacting transcription factors are many that are regulated by other signaling pathways, adding to the complex and integrated nature of the regulation of different cellular responses [225].

### 8.1. Cross-Talk at the Receptor Level

One of the most important cross-talks is with the integrin pathway. It has been described that TGF-β induces the expression of the α_v_β_3_ integrin and also enhances its basal weak association with TGFβRII in fibroblasts. Interestingly, even though TGF-β alone does not significantly affect fibroblast proliferation, the co-exposure of fibroblasts to TGF-β and integrin ligands such as vitronectin (VN), significantly enhances their proliferation [226]. The elucidation of this mechanism gave more insight into the contribution of TGF-β to fibrotic disorders. Whether such a mechanism also coordinates with the mechanism of activation of latent TGF-β (as described in Section 2), remains to be explained.

Activation of the insulin pathway enhances the physical interaction between AKT and SMAD3, which prevents SMAD3 phosphorylation thus leading to inhibition of SMAD3-mediated transcription and attenuation of TGF-β signaling [227,228]. Serine-threonine kinase receptor-associated protein (STRAP) acts as a mediator in the cross-talk between TGF-β and PI3K/PDK1 signaling. Insulin-induced association of STRAP with PDK1, leads to PDK1 activation and subsequent AKT phosphorylation, which finally results in inhibition of TGF-β signaling [229].

Neurotrophin-3 receptor (NTRK3 or TrkC) and its related chimeric constitutively active ETV6-NTRK3, both interact with the TGF-β type II receptor to prevent complex formation between TGFβRII and TGFβRI, thus suppressing TGF-β signaling [230,231].

Neuropilin-1 is a transmembrane receptor that positively regulates TGF-β signaling as it interacts with the TGF-β type II receptor and enhances TGF-β-induced SMAD2/3 phosphorylation in fibroblasts [232].

### 8.2. Cross-Talk at the Level of SMAD (or Other Signaling Effector) Activation

A cross-talk that couples TGF-β signaling to the Hippo pathway has been described whereby the Hippo pathway effectors Yes-associated protein/transcriptional coactivator with PDZ-binding motif (YAP/TAZ) interfere with the SMAD dependent signaling. The Hippo pathway senses cell density in order to control tissue growth by regulating the subcellular localization of the transcriptional regulators YAP/TAZ. High cell density leads to the assembly of the polarity complex Crumbs that interacts with YAP/TAZ, and results in the phosphorylation and cytoplasmic retention of YAP/TAZ. YAP/TAZ in turn associate with active SMAD2/3 complexes and sequester them to the cytoplasm, thus suppressing TGF-β signaling. [233]. The nuclear accumulation of YAP/TAZ overcomes TGF-β induced cell cycle arrest and together with SMADs promotes a pro-tumorigenic transcriptional program in breast cancer cells [234]. The YAP/TAZ–TGF-β/SMAD crosstalk has an important role also in fibrogenesis as increased ECM stiffness promotes TGF-β induced SMAD signaling via the mechanical regulation of YAP/TAZ activity [235]. Moreover, it has also been described that YAP/TAZ over-activation as a result of loss of the upstream Lats1/2 kinases in primary hepatoblasts, leads to enhanced TGF-β signaling, as YAP directly regulates *Tgfb2* transcription [236]. Interestingly, cell density regulates TGF-β signaling also independently from Hippo pathway activation, but this time at the level of TGF-β receptors. Specifically in epithelial polarized cells, TGF-β receptors are re-distributed exclusively at the basolateral cell surface, an event that prevents apically delivered TGF-β from binding to the receptors at the apical compartment and thus negatively regulates TGF-β signaling [237].

TRAP-1-like protein (TLP) is an adaptor protein that is known to interact with the TGF-β type II receptor, but has been also described to interact with the common mediator SMAD4 in response to TGF-β. Via this interaction, TLP inhibits specifically the formation of SMAD3/4 complexes, thus shifting the balance towards TGF-β/SMAD2 transcription [238].

Interestingly, km23-1 that was earlier mentioned for its role in the regulation of SMAD2 subcellular localization has also been described to be important for the TGF-β-dependent induction of Ras/ERK/JNK pathways by acting as an adaptor molecule between the TGF-β receptor and Ras [119]. The examples of signaling cross-talk are many and constantly increase as different physiological or pathophysiological scenarios become uncovered in diverse cell types.

## 9. Future Perspectives and Concluding Remarks

TGF-β represents a major signaling network that permeates the function of many biological processes during embryonic and newborn development, adult homeostasis and disease onset and progression. The signaling pathways that make part of this network have been uncovered to a large extent and continuous efforts bring new exciting regulatory mechanisms as “unexpected” sides of the versatility and multifaceted regulation of the network (Figure 3 and Figure 6).

Future research will most certainly illuminate new such regulatory mechanisms and will clarify a number of outstanding open areas in this field. At the level of extracellular ligand synthesis and activation-presentation to signaling receptors, a major current open area is the question of homodimeric and heterodimeric TGF-β family members. In other words, can all biological actions of the TGF-βs be explained by the well-established homodimeric ligands or can certain aspects, and which, be explained by heterodimeric (TGF-β1/2, TGF-β1/3, or even TGF-β2/3) ligands. This model is currently promoted by in vivo studies of the action of BMP family members, as presented earlier. The mechanism of ligand activation from its latent form currently appears with some controversial statements, which often downplay the role of metalloproteases and emphasize only the function of integrin receptors. A more unbiased and possibly biological system-wide analysis of the relative contribution of different molecular mechanisms that activate the TGF-βs in various tissues and during various pathophysiological conditions is needed, and will provide better understanding for the biological circumstances that activate these multipotent growth factors.

As illustrated in this article, work at the level of the TGF-β receptors and their regulation awaits completion in the near future. The TGF-β field continuously widens thanks to the serendipitous findings originating from a diverse array of pathophysiological conditions that are genetically or molecularly analyzed and conclude that molecules implicated in the transmission or control of TGF-β signaling, play important roles in the specific disease. Based on this, more detailed molecular links between the signaling mechanisms and the pathophysiology of various diseases is also another area that awaits refinement and will possibly contribute to new ideas for the diagnosis and therapy of these diseases. Two specific aspects of modern TGF-β research are relevant to discuss here. a) An important toolbox that is missing from the analysis of the TGF-β pathway in physiological embryogenesis and in the analysis of pathogenetic mechanisms is antibodies that can monitor the very early steps of signal transduction. These include ligand-receptor complexes and receptor kinase activation and phosphorylation. Currently, only phospho-SMAD antibodies are used to measure TGF-β signaling processes in embryonic or adult tissues or even tissues from disease models in humans or other animals. As explained in the article, SMAD phosphorylation lies downstream from the ligand-receptor complex and is regulated by multiple proteins, leading to only a rough estimate of signaling activity. b) The so-called non-SMAD signaling pathways are essentially other pathways (e.g., MAP kinases, Rho GTPases, PI3 kinase etc), which get activated by a large variety of growth factors, cytokines, chemokines, and stress factors. For this reason, the relative contribution of these signaling molecules to TGF-β responses, independent from parallel responses to other cytokines (cross-talking pathways) is essentially impossible. Research aiming at identifying post-translational modifications of the various signaling molecules that might get activated only during TGF-β signaling would provide a unique tool that could establish the true importance of the alternative, non-SMAD signaling mediators downstream of TGF-β pathways in vitro and in vivo.

Additional major fronts of research in the TGF-β field are the emerging functions of SMAD signaling in mediating chromatin-based (epigenetic) and RNA-based (including non-coding RNA) cell biological mechanisms. The transport of TGF-β and TGF-β signaling components via exosomes has been introduced to the field but is difficult to reconcile when the same growth factor is also deposited to the extracellular microenvironment. What is the need of extracellular vesicle-mediated communication of signaling components, when the pathway initiates by extracellularly deposited proteins? Can this action of extracellular vesicles be understood based on the ability of TGF-β signaling molecules, e.g., SMADs to interact with RNAs or other chromatin components that are then packaged into vesicles for delivery and signal propagation in a paracrine manner? These and additional scientific questions are constantly generated in the prolific field of TGF-β signal transduction and generate a rich terrain for future investigation and discovery that permeates all aspects of multicellular organismic life and disease.

## Figures and Tables

**Figure 1 biomolecules-10-00487-f001:**
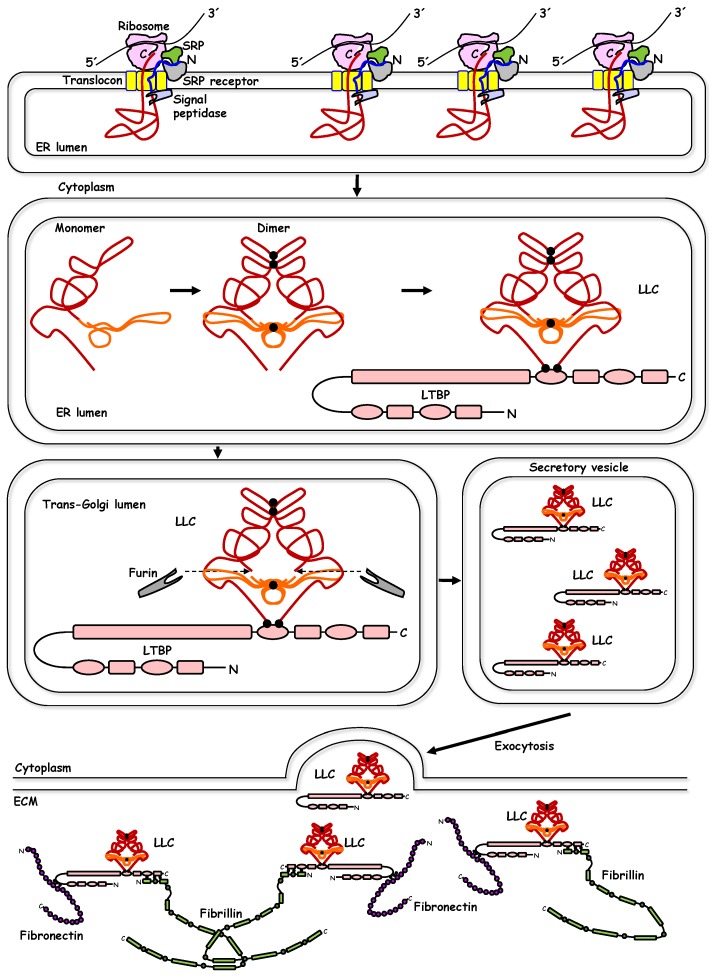
Biosynthesis and extracellular deposition of transforming growth factor-β (TGF-β). A sequence of biochemical events is shown from the top left to the bottom, guided by black arrows. Ribosomes attached to the endoplasmic reticulum (ER) translate the TGF-β mRNA (black line 5′-3’) into TGF-β protein (red line with blue signal peptide). The signal peptide associates with the signal recognition protein (SRP), which associates with the SRP receptor and the translated polypeptide is transported through the translocon channel into the lumen of the ER where the signal peptidase cleaves the signal peptide, generating a pro-TGF-β monomer that folds in the lumen of the ER (red polypeptide corresponds to the N-terminal long polypeptide known as latency associated peptide (LAP) and orange polypeptide correspond to the mature C-terminal polypeptide). The architecture of pro-TGF-β follows the crystallographic structure of the molecule. Dimerization of pro-TGF-β takes place in the ER lumen via three disulfide bonds (black dots), two in the prodomain and one in the mature domain. The dimeric pro-TGF-β crosslinks via disulfide bonds (black dots) to the latent TGF-β binding protein (LTBP) in the ER lumen, forming the large latent complex (LLC). The LLC translocates from the ER lumen to the cis- (not shown) and then to the trans-Golgi cisternae. For simplicity, the prodomain N-linked glycosylation is not shown. In the trans-Golgi, furin protease cleaves at the junction of the prodomain with the mature domain (dotted arrows). The cleaved LLC accumulates in secretory vesicles that undergo exocytosis and secrete the LLC to the extracellular environment where the LLC incorporates into the matrix (ECM). LLC crosslinking to fibrillin (three disulfide bonds, black dots) and to fibronectin (three more disulfide bonds, black dots) is shown. All relevant proteins are drawn emphasizing their domain architecture, without clarifying the identity of each domain.

**Figure 2 biomolecules-10-00487-f002:**
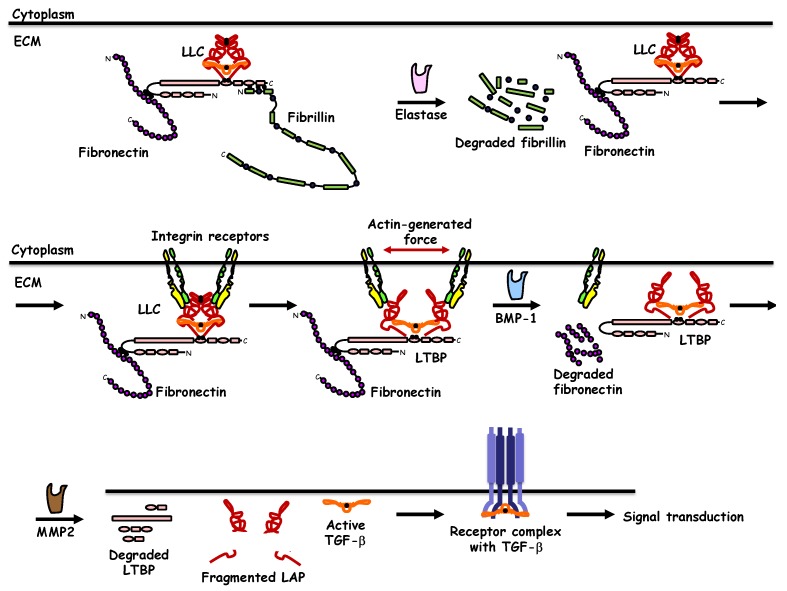
Activation of latent TGF-β. A sequence of biochemical events is shown from the top left to the bottom right, guided by black arrows. The large latent complex of TGF-β (LLC) deposited to the ECM via crosslinking to fibronectin and fibrillin is shown. Elastase proteolytically cleaves fibrillin. Integrin receptors on the plasma membrane associate with the RGD peptides (not shown) of the TGF-β prodomain. Integrin heterodimers of α- and β- integrin chains are shown in different color. Integrins, via their association to the actin cytoskeleton (not shown) exert force and change the conformation of the LLC prodomain, initiating the mature TGF-β release process. BMP-1 proteolytically cleaves fibronectin and MMP2 cleaves LTBP and the prodomain of TGF-β, generating fragmented prodomain, i.e., latency associated peptide (LAP), and releasing mature TGF-β. Active TGF-β associates with the signaling type II and type I receptors and initiates signal transduction. The role of coreceptors in ligand presentation is not shown.

**Figure 3 biomolecules-10-00487-f003:**
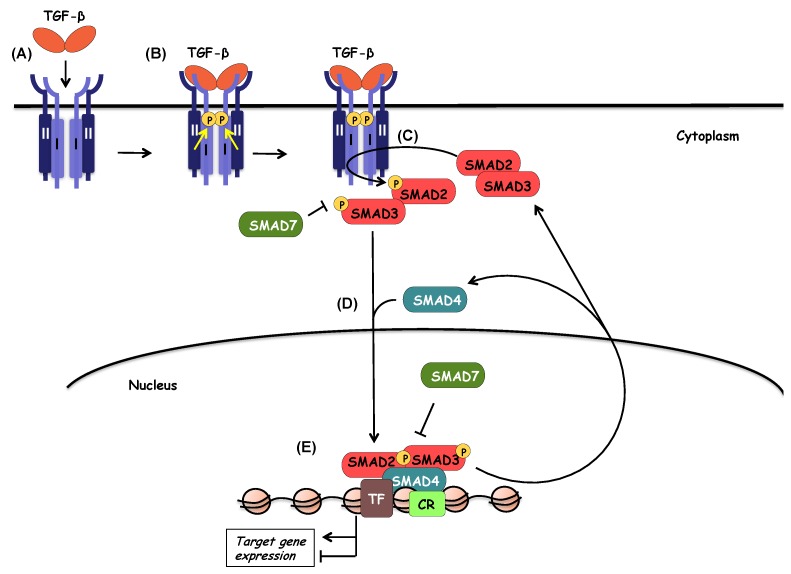
The TGF-β/SMAD signaling pathway. During the first steps of TGF-β signaling, TGF-β ligand binds to a heteromeric complex of type II, and type I receptors (**A**). Upon ligand binding, type II receptor phosphorylates and activates type I receptor (**B**). Activated type I receptor in turn phosphorylates and activates the receptor-activated SMADs (R-SMADs), SMAD2 and SMAD3 (**C**). SMAD7 competes with R-SMADs for interacting with type I receptor, thus preventing R-SMAD activation and proper propagation of the signaling. Activated R-SMADs dissociate from type I receptors in order to form a complex with the common mediator SMAD4 (**D**). The trimeric complex translocates to the nucleus where it associates with high-affinity DNA binding transcription factors (TF) and chromatin remodeling proteins (CR) in order to positively or negatively regulate the transcription of target genes (**E**). SMAD7 can also inhibit the transcriptional activity of the nuclear SMAD complex.

**Figure 4 biomolecules-10-00487-f004:**
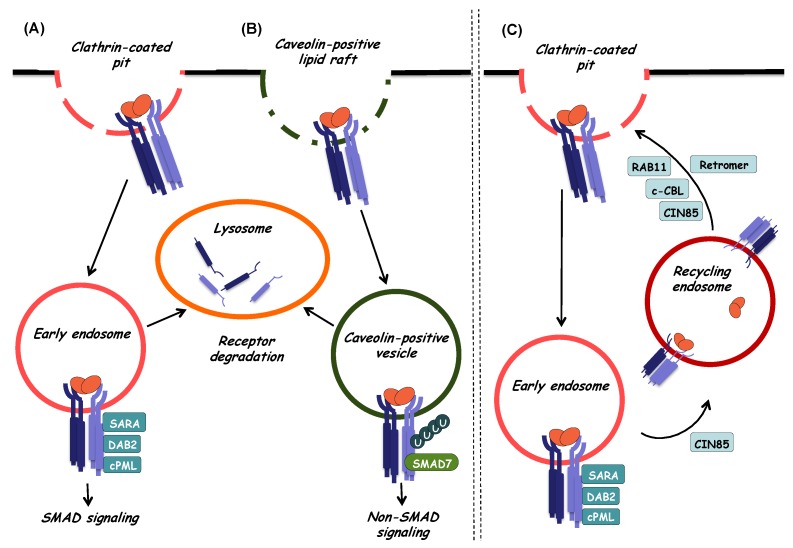
Internalization and intracellular sorting of TGF-β receptors. Endocytosis and intracellular sorting of TGF-β receptors play an important role in the regulation of TGF-β signaling outcome and can be mediated via the two major endocytic pathways. (**A**) SMAD-dependent TGF-β signaling can initiate at the cell surface in clathrin-coated pits. When receptors internalize via clathrin-coated vesicles, they are directed to early endosomes. In these early stages of endocytosis, association of TGF-β receptors with SARA, cPML, and Dab-2 adaptor proteins, leads to the enhancement of TGF-β-induced SMAD activation and subsequent propagation of SMAD-dependent signaling. Internalized TGF-β receptors found in early endosomes can be sorted for degradation. In this case, they enter late endosomes (not shown) and finally reach lysosomes where degradation takes place. (**B**) Internalization of TGF-β receptors can also take place at caveolin-positive lipid raft compartments on the cell membrane, and in this case, internalized receptors enter caveolin-positive vesicles. There, the TGF-β type I receptor preferentially associates with SMAD7, which can control receptor turnover via recruitment of ubiquitin ligases and deubiquitylating enzymes thus regulating ubiquitylation and subsequent lysosomal degradation of receptors. Internalization of TGF-β receptors by lipid raft/caveolar-mediated endocytosis can also promote non-SMAD TGF-β signaling as SMAD7 competes with SMAD2/3 for interaction with the TGF-β type I receptor. (**C**) Once internalized via clathrin-coated pits, TGF-β receptors enter early endosomes. From there, receptors can be sorted to recycling endosomes in order to return back to the cell surface where they can respond again to ligand stimulation. The small GTPase RAB11, the adaptor protein CIN85, the ubiquitin ligase c-CBL and the retromer complex can facilitate recycling in different cell types.

**Figure 5 biomolecules-10-00487-f005:**
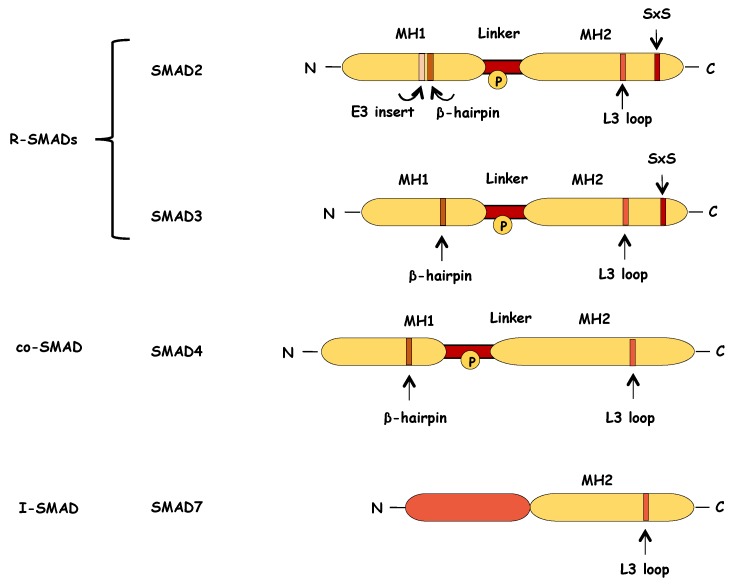
Schematic representation of structure of the SMAD proteins. R-SMAD and SMAD4 proteins consist of two highly conserved domains, the MH1 and MH2 domains, which are separated by a non-conserved linker region. The N-terminal MH1 domain of R-SMADs contains a β-hairpin structure that is critical for DNA binding. In the case of SMAD2, the MH1 domain contains also an extra amino-acid sequence (E3 insert) that negatively regulates the DNA binding capacity of SMAD2. The C-terminal MH2 domain of R-SMADs contains an L3 loop that mediates the interaction between R-SMADs and the activated type I receptor. This L3 loop is also part of the SMAD4 structure and it is important for the formation of SMAD trimeric complexes. At their very C-terminus, R-SMADs, have a short conserved motif of two serines separated by one amino acid (Ser-X-Ser (SXS)) that are phosphorylated by the activated type I receptor, thus leading to the R-SMAD activation. The linker region encompasses multiple phosphorylation sites (P in circle), and it is targeted by various kinases that modulate SMAD stability and function. SMAD7, the inhibitory SMAD, retains the conserved MH2 domain but lacks the SXS motif at the C-terminus and the N-terminal region presents small similarity to the MH1 domain.

**Figure 6 biomolecules-10-00487-f006:**
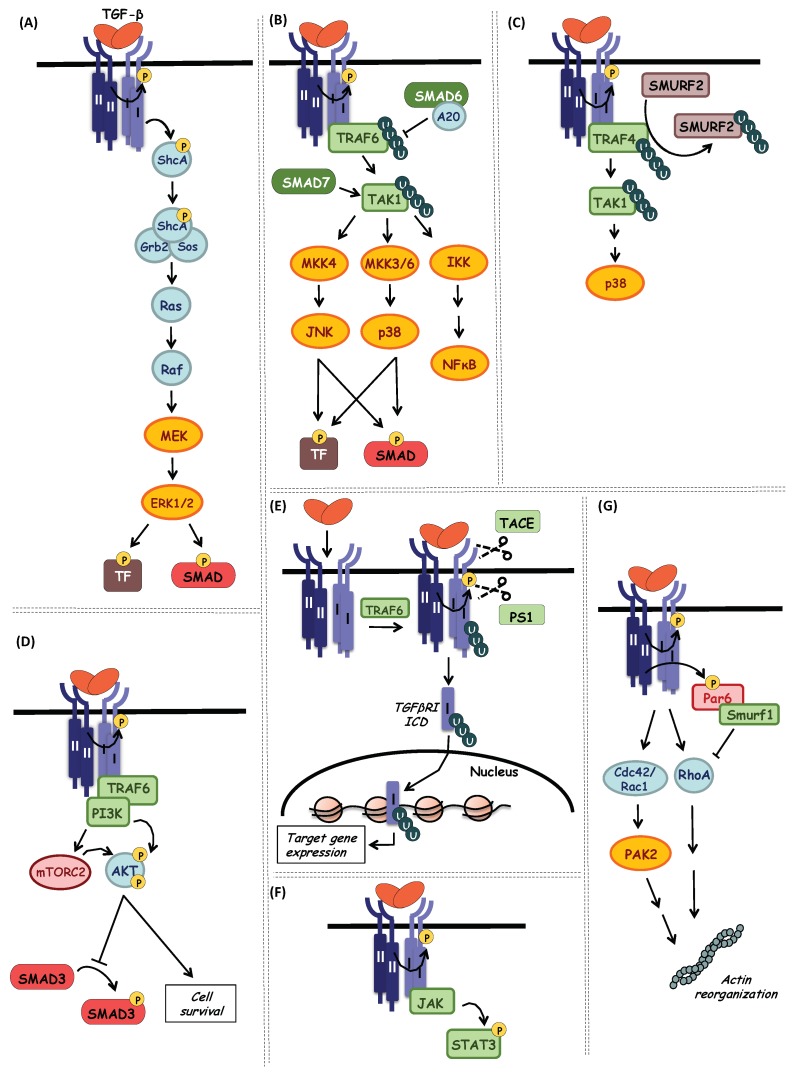
The TGF-β non-SMAD signaling pathways. (**A**) The ERK-MAP kinase pathway. First step for the activation of the TGF-β-induced ERK pathway is the phosphorylation of ShcA by the activated type I receptor. TGF-β-induced tyrosine phosphorylation of ShcA promotes formation of ShcA/Grb2/Sos complex and Ras activation. This leads to the sequential activation of Raf, MEK1/2 and finally ERK1/2. Activated ERK1/2 phosphorylate transcription factors (TF), thus contributing to TGF-β-induced transcriptional responses. Activated ERK1/2 can also phosphorylate SMADs at the linker region to regulate their activity. (**B**) The p38/JNK and NF-κB pathway (via TAK1). Upon ligand binding, TGF-β receptor complexes interact with TRAF6 promoting its autoubiquitylation. SMAD6 can inhibit TRAF6 ubiquitylation and activation by recruiting the deubiquitylating enzyme A20. TRAF6 activates TAK1 via Lys63-linked polyubiquitylation and activated TAK1 in turn activates through phosphorylation MAP kinase kinases (MKKs) MKK4, MKK3 and MKK6. MKKs activate their downstream kinases JNK and p38, which can then phosphorylate their target transcription factors (TF) in order to regulate transcription. SMAD7 enhances the activation of the p38 pathway as it acts as a scaffolding protein for TAK1, MKK3 and p38. Activated JNK and p38 phosphorylate also SMADs at the linker region, thus regulating SMAD-dependent transcriptional responses as well. Finally, TAK1 activates also IKK, which eventually leads to the activation of NF-κB signaling. (**C**) The PI3K/AKT/mTOR pathway. TGF-β promotes PI3K/AKT activation via direct interaction of the p85 subunit of PI3K (not shown) with TGF-β receptors. TGF-β-induced autoubiquitylation of TRAF6 results in recruitment and phosphorylation of AKT. TGF-β via PI3K, promotes also activation of mTORC2, which in turn can also phosphorylate and activate AKT promoting cell survival. Moreover, activated AKT prevents phosphorylation of SMAD3, thus attenuating SMAD3-dependent signaling. (**D**) TGF-β signaling by type I receptor intracellular domain signaling. The transmembrane metalloprotease TACE, promotes ectodomain cleavage of type I receptor, which is then followed by TRAF6-mediated ubiquitylation of the cytoplasmic domain of type I receptor and recruitment of presenilin-1 (PS1), part of the γ-secretase complex. PS1 proteolytically cleaves the ubiquitylated intracellular cytoplasmic domain of the receptor (TGFβRI ICD), which is released into the cytoplasm. Then, TGFβRI ICD translocates to the nucleus where it associates with other co-factors (not shown) and induces the expression of target genes. (**E**) MAP kinase pathway activation via TRAF4. The MAP kinase pathway can also be activated via TRAF4, another E3 ligase that upon ligand binding is recruited to the receptor complex, gets autoubiquitylated and then activates TAK1 via polyubiquitylation, eventually leading to activation of the p38 pathway. At the same time, TRAF4 targets SMURF2 for polyubiquitylation and subsequent degradation, thus contributing to the stability of TGF-β type I receptor. (**F**) The JAK-STAT pathway. STAT3 gets phosphorylated and activated by JAK (which interacts with the type I receptor) in response to TGF-β in order to regulate the expression of subset of TGF-β target genes. (**G**) The Rho-(like) GTPase pathway activation. TGF-β induces activation of RhoA GTPase (via both SMAD-independent and SMAD-dependent mechanisms), which eventually results in actin cytoskeleton reorganization and formation of stress fibers. Additionally, Par6, a regulator of cell polarity, once phosphorylated by TGF-β type II receptor, recruits Smurf1 E3 ligase that targets RhoA for degradation, eventually leading to tight junction dissociation. Upon TGF-β stimulation, Rho-like proteins Cdc42 and Rac1 are also activated and promote actin reorganization via activation of PAK2.

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
