# Peer review of "TGF-β Signaling"

_biomolecules, 2020, doi:10.3390/biom10030487_

Round 1
Reviewer 1 Report
This review identifies key elements of TGF-beta pathway (i.e., ligand receptor interactions, receptor activation and complex assembly, mechanisms of receptor internalization and degradation and SMAD and non-SMAD cascades) critically involved in signal transduction leading to genetic responses. The authors present a well-written and a comprehensive overview TGF-beta pathway citing critical literature. The following minor modifications would further add to the scientific value to the revised manuscript, which will be suitable for publication.
1. Some of the non-SMAD pathways activated by TGF-beta such as YAP/TAZ and beta-catenin should be described.
2. An assessment of where TGF-beta field is heading and the critical areas of TGF-beta signaling which require further exploration need to included.
3. It would be useful to include what cells or systems are used for key studies given the cell type specific differences in TGF-beta signaling.
Author Response
Reviewer 1:
This review identifies key elements of TGF-beta pathway (i.e., ligand receptor interactions, receptor activation and complex assembly, mechanisms of receptor internalization and degradation and SMAD and non-SMAD cascades) critically involved in signal transduction leading to genetic responses. The authors present a well-written and a comprehensive overview TGF-beta pathway citing critical literature. The following minor modifications would further add to the scientific value to the revised manuscript, which will be suitable for publication.
Author response:
We thank this reviewer for the positive words and for finding this article comprehensive.
- Some of the non-SMAD pathways activated by TGF-beta such as YAP/TAZ and beta-catenin should be described.
Author response:
We appreciate this comment and in response we have added a new paragraph on TGF-β and Hippo pathway crosstalk in section 8.2, lines 1251-1273.
- An assessment of where TGF-beta field is heading and the critical areas of TGF-beta signaling which require further exploration need to included.
Author response:
This is an important comment and responding to this we have changed the last section 9. Concluding remarks to 9. Future perspectives and concluding remarks. This section has been enhanced significantly by pointing to several open areas for current and future research, thus attempting to presents our views on “where the TGF-β field is heading toward”. The new text can be found in section 9, lines 1289-1352.
- It would be useful to include what cells or systems are used for key studies given the cell type specific differences in TGF-beta signaling.
Author response:
This is an important comment. In the previous version, we included the information on biological models as much as possible in the section of non-SMAD signaling (section 7). We have now responded to this comment by adding new text also in the section of SMAD signaling, section lines 670-681, where key evidence for the biological role of SMADs is presented and a general overview of the model systems used for establishing the biological roles of SMAD signaling is presented. We have also enhanced the detail in the non-SMAD signaling section 7, lines 1012-1157.
Reviewer 2 Report
First of all, I want to congratulate the authors on a very comprehensive and nicely written review. This review aims to give an overview of TGF-β signaling with focus on the TGF-βs themselves. They touch upon many important aspects of TGF-β signaling, from synthesis and maturation of ligands, how receptors and co-receptors work and are regulated, different signaling outcomes (SMAD- and non-SMAD signaling as well as signaling cross-talk). Many of these aspects are quite complex, but the authors have presented the information in a way that is understandable.
I have a few minor comments:
- The scope and limits of the review are cleary stated in the introduction. I would have preferred a hint in the abstract that the review would only focus on TGF-βs and not the other family members.
- The language is very good and the disposition of the content is logical. The figures that support the text are also clear.
- In section 2, the synthesis of TGF-βs are described in detail. It could be even easier to comprehend/visualize this part if the authors included a figure to show different stages of synthesis, cleavage etc.
- Figure 1 is placed at a natural place in the text, but Figures 2 and 3 should be moved to the place in the text where they are mentioned. The resolution of the figures in the paper seems to low, but the supplementary file is OK.
- Figure 2: CIN85 is described in the text and maybe it should be included in the figure as well. Likewise, cPML is in the figure, but its function is not described in the text, as it should be.
- In section 4, the authors describe the function of co-receptors. This is very important and interesting. I miss the mentioning of soluble endoglin and how this works. Indeed, a publication in PNAS last year (Lawera et al.) sheds new light on the function of soluble endoglin and this could have been discussed in the review.
- Section 6.1 describes the SMADs’ structure and domains. Also in these case, I think the review would benefit from a figure that aligns the SMADs and show their respective domains.
- In section 6.4 about posttranslational modification of SMADs, the authors discuss the regulatory funcion of phosphorylation in the linker region of SMADs. Although I understand that there is not enough space to include everything, I miss information about regulatory phosphorylation of the N-terminus, such as Thr8.
- The «β» has disappeared in the start of the abstract and in headline for section 3.
- In the abstract, would it be more correct to say «developmental stage-dependent» instead of «developmental type-dependent»?
- In page 2, line 59, I would have changed «all» to «most», since there are exceptions to the rule for how proteins are secreted.
- In page 3, line 98, I would have written the whole full term for NMR, even if this is a commonly used abbreviation.
- Same as above, page 3, line 124: Please write what RGD is an abbreviation of.
- In page 13, line 523, I believe that the correct term is «Hepatocyte growth factor ...» instead of «Hepatic growth factor ...». Please check this.
- In page 13, line 540, I would have written «may result» instead of «results», as the formation of mixed R-SMAD complexes may not happen all the time.
Overall, I am very satisfied with the quality of this review and think it provides a very nice and clear introduction to TGF-β signaling, not only to people new to this pathway, but also to scientists that are already familiar with TGF-β signaling.
Author Response
Reviewer 2:
First of all, I want to congratulate the authors on a very comprehensive and nicely written review. This review aims to give an overview of TGF-β signaling with focus on the TGF-βs themselves. They touch upon many important aspects of TGF-β signaling, from synthesis and maturation of ligands, how receptors and co-receptors work and are regulated, different signaling outcomes (SMAD- and non-SMAD signaling as well as signaling cross-talk). Many of these aspects are quite complex, but the authors have presented the information in a way that is understandable.
Author response:
We thank this reviewer for the positive comments and for finding this article understandable.
I have a few minor comments:
- The scope and limits of the review are cleary stated in the introduction. I would have preferred a hint in the abstract that the review would only focus on TGF-βs and not the other family members.
Author response:
In response to this comment, the new abstract, line 23, has an extra sentence clarifying that the article mainly focuses on the prototype TGF-β, and not on every one of the 33 members of the family.
- The language is very good and the disposition of the content is logical. The figures that support the text are also clear.
Author response:
We appreciate all the positive comments by the reviewer.
- In section 2, the synthesis of TGF-βs are described in detail. It could be even easier to comprehend/visualize this part if the authors included a figure to show different stages of synthesis, cleavage etc.
Author response:
We appreciated this comment and in response we have added two new figures, Figure 1 and 2, the first on TGF-β biosynthesis and the second on TGF-β activation from its latent form. The two new figures have been placed into the document file and corresponding figure legends have been added in lines 107-132 (Figure 1) and lines 185-199 (Figure 2).
- Figure 1 is placed at a natural place in the text, but Figures 2 and 3 should be moved to the place in the text where they are mentioned. The resolution of the figures in the paper seems to low, but the supplementary file is OK.
Author response:
We understand the comment. The figures were placed by the journal and we have now attempted to place all 6 figures in the best possible placement within the document. We understand that the journal may change this format later.
- Figure 2: CIN85 is described in the text and maybe it should be included in the figure as well. Likewise, cPML is in the figure, but its function is not described in the text, as it should be.
Author response:
In agreement with the reviewer, new Figure 4, panel C indicates the function of CIN85 in receptor recycling.
- In section 4, the authors describe the function of co-receptors. This is very important and interesting. I miss the mentioning of soluble endoglin and how this works. Indeed, a publication in PNAS last year (Lawera et al.) sheds new light on the function of soluble endoglin and this could have been discussed in the review.
Author response:
In agreement with the reviewer, we have added new text on the soluble form of endoglin in section 4 on coreceptors, lines 485-499.
- Section 6.1 describes the SMADs’ structure and domains. Also in these case, I think the review would benefit from a figure that aligns the SMADs and show their respective domains.
Author response:
In response to this comment we have generated new Figure 5 that illustrates the three classes of SMAD proteins, their domains and certain important structural features.
- In section 6.4 about posttranslational modification of SMADs, the authors discuss the regulatory funcion of phosphorylation in the linker region of SMADs. Although I understand that there is not enough space to include everything, I miss information about regulatory phosphorylation of the N-terminus, such as Thr8.
Author response:
We appreciated this comment by the reviewer and we have responded by adding new text, covering many phosphorylation events that span the three domains of SMADs, which are all included in section 6.4, lines 798-828.
- The «β» has disappeared in the start of the abstract and in headline for section 3.
Author response:
We thank the reviewer for this meticulous comment. The typographical errors have now been corrected, lines 9 and 200.
- In the abstract, would it be more correct to say «developmental stage-dependent» instead of «developmental type-dependent»?
Author response:
In agreement with the reviewer this change has been implemented in line 13.
- In page 2, line 59, I would have changed «all» to «most», since there are exceptions to the rule for how proteins are secreted.
Author response:
In agreement with the reviewer this change has been implemented in new line 66.
- In page 3, line 98, I would have written the whole full term for NMR, even if this is a commonly used abbreviation.
Author response:
In agreement with the reviewer this change has been implemented by spelling out NMR as nuclear magnetic resonance in new line 134. Since this term is not used more than once in the article the abbreviation NMR has been eliminated.
- Same as above, page 3, line 124: Please write what RGD is an abbreviation of.
Author response:
In agreement with this comment we have clarified the identify of RGD by explaining as follows in line 161-162: Arg, Gly, Asp (RGD) tri-peptide motif.
- In page 13, line 523, I believe that the correct term is «Hepatocyte growth factor ...» instead of «Hepatic growth factor ...». Please check this.
Author response:
In agreement with this comment we have made this important correction for the name of HGF in line 752.
- In page 13, line 540, I would have written «may result» instead of «results», as the formation of mixed R-SMAD complexes may not happen all the time.
Author response:
We thank the reviewer for this important comment, which we have implemented in line 766.
Overall, I am very satisfied with the quality of this review and think it provides a very nice and clear introduction to TGF-β signaling, not only to people new to this pathway, but also to scientists that are already familiar with TGF-β signaling.
Author response:
We thank this reviewer for all the positive comments.